# Bottleneck-Guided Spectral Subgoals for Offline Goal-Conditioned RL

**Hebin Liang** [* 1]  **Yi Ma** [* 2]  **Chenjun Xiao** [3]  **Zibin Dong** [1]  **Zilin Cao** [1]  **Fei Ni** [4]  **Yifu Yuan** [1]  **Jianye Hao** [1]

## Abstract

Offline goal-conditioned RL (OGCRL) learns to reach arbitrary goals from the offline dataset, but long-horizon performance hinges on crossing a handful of hard-to-cross bottlenecks. These bottlenecks not only dictate the feasible paths toward the goal but also act as critical keypoints, marking the transitions between adjacent regions and providing the agent with essential directional guidance. Prior hierarchical methods pick subgoals by time or short-horizon value heuristics, which do not explicitly localize the bottleneck, and therefore the agent loses the clear guidance that bottlenecks could provide about where to pass next. We instead model long-horizon planning as "cross the next bottleneck": we apply Laplacian spectral clustering to the offline dataset to expose bottlenecks, identify trajectories from the offline dataset that cross these boundaries, and use the corresponding boundary-supported states as keypoint (KP) candidates. The most representative KPs are then instantiated from these candidates, and a directed KP reachability graph $\mathcal{G}_{\mathrm{KP}}$ is constructed based on the resulting KPs. We then restrict high-level choices to these bottleneck states and use a pluggable low-level controller to execute the short transitions between them. We provide theory showing that under a standard metastable decomposition of the state space, routing through bottlenecks yields an (approximately) optimal one-step subgoal in terms of hitting time, and that Laplacian spectra recover bottlenecks with high overlap. Thus, Laplacian spectral clustering can discover near-optimal subgoals under these assumptions. Empirically, the same pattern holds: across D4RL and OGBench, our method achieves state-of-the-

art results on a broad set of navigation and manipulation tasks and across diverse dataset regimes, for example, **96.5%** on **AntMaze** and **84.7%** on **Franka-Kitchen**.

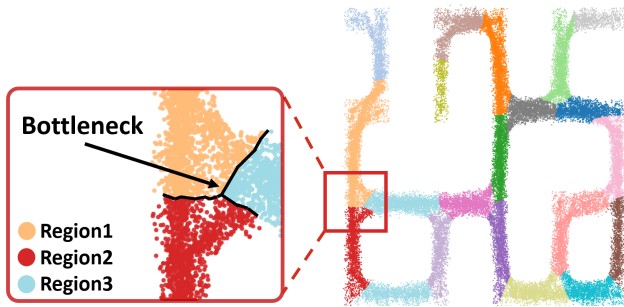

*Figure 1.* Laplacian spectral clustering in `pointmaze-large`. Colors indicate regions, and boundaries visually align with hard-to-cross bottlenecks.

## 1. Introduction

Long-horizon sparse rewards remain a core challenge for offline goal-conditioned reinforcement learning (OGCRL): datasets are limited and biased, online interaction is unavailable, and credit assignment couples with planning over long time scales. In most OGCRL tasks, the state space decomposes into well-connected regions (e.g., rooms and corridors in navigation) linked by a few hard-to-cross bottlenecks (e.g., doorways or narrow chokepoints in mazes). These bottlenecks act as structural keypoints that any successful trajectory must pass and thus provide a clear high-level guidance signal: cross the next bottleneck to move from one region to the next. Fig. 1 illustrates our method's identification of regions and bottlenecks: each colored patch corresponds to a region, and the boundaries between patches visually align with bottlenecks.

Existing long-horizon GCRL and hierarchical/offline planning methods often introduce intermediate targets, graph waypoints, skills, or finite-horizon trajectory plans. Imagined subgoals are predicted approximately halfway to the goal and used to regularize policy learning rather than as explicit test-time waypoints (Chane-Sane et al., 2021). Adjacency-constrained HRL restricts candidate subgoals to

---

[*]Equal contribution [1]Tianjin University, Tianjin, China [2]Shanxi University, Taiyuan, China [3]The Chinese University of Hong Kong, Shenzhen, Shenzhen, China [4]Imperial College London, London, United Kingdom. Correspondence to: Chenjun Xiao <chenjunx@cuhk.edu.cn>, Jianye Hao <jianye.hao@tju.edu.cn>.

*Proceedings of the 43rd International Conference on Machine Learning*, Seoul, South Korea. PMLR 306, 2026. Copyright 2026 by the author(s).

a $k$-step neighborhood to preserve local feasibility (Zhang et al., 2020), while graph-based GCRL methods use planned subgoal sequences to guide or distill goal-conditioned policies (Kim et al., 2023). Related offline skill and trajectory-planning methods learn reusable primitives or generate finite-horizon trajectories with diffusion-style planners (Ajay et al., 2020; Pertsch et al., 2021; Janner et al., 2022; Ajay et al., 2022). Recent ESD further considers locally reachable states as elastic subgoal candidates (Zhang et al., 2025). Despite these differences, such choices are typically local, finite-horizon, or value-driven rather than explicitly bottleneck-aligned, and thus may miss the structural guidance signal provided by bottlenecks: where the agent must pass next.

To address these issues, we advance a simple principle for hierarchical OGCRL: under a metastable decomposition, a near-optimal nontrivial one-step subgoal is located at the next bottleneck.

To identify bottlenecks from offline data, we first learn a Laplacian representation $\phi_\theta$ from replay so that the state representation varies slowly within well-connected regions while varying sharply near low-conductance boundaries. Spectral clustering in this embedding yields region labels. Fig. 1 shows the resulting partition on `pointmaze-large`, where boundaries between colors coincide with hard-to-cross bottlenecks. We then identify trajectories from the offline dataset that cross these boundaries and use the corresponding boundary-supported states as KP candidates. The most representative KPs are then instantiated from these candidates, and a directed KP reachability graph $\mathcal{G}_{\mathrm{KP}}$ is constructed based on the resulting KPs. At deployment, we restrict high-level choices to these KPs and use a pluggable low-level controller (e.g., Decision Diffuser or a lightweight MLP) to execute the short transitions between successive KPs. We name our method **BASS** (**B**ottleneck-**A**ware **S**pectral **S**ubgoaling).

**Contributions.** Our contributions form an integrated framework: (1) We introduce a bottleneck-guided criterion that ties subgoal selection to the next bottleneck, underpinned by theoretical analysis. (2) We develop a keypoint discovery method based on Laplacian spectral clustering to automatically derive candidate bottleneck keypoints from the offline datasets. (3) We design a hierarchical algorithm for OGCRL that plans by routing through these keypoints using a pluggable low-level controller. (4) Extensive experiments on diverse navigation and manipulation benchmarks from D4RL (Fu et al., 2020) and OGBench (Park et al., 2025) demonstrate consistent bottleneck recovery and performance gains across varied data regimes.

## 2. Preliminaries

**OGCRL and metastable bottlenecks.** We study offline goal-conditioned RL (OGCRL) where a policy $\pi(a \mid s, g)$ is learned from a fixed replay dataset. Long horizons with sparse rewards make progress hinge on planning rather than short-term value. In practice, most OGCRL tasks are *metastable*: the state space decomposes into regions that are easy to traverse, i.e., fast intra-region mixing, and these regions are connected only through a few hard-to-cross bottlenecks that are rarely crossed under the dataset-induced dynamics. Consequently, success to distant goals is governed by *whether the agent crosses the next bottleneck* rather than by taking a few more steps within the current region (see Fig. 1).

**Laplacian RL: what it is and why we use it.** Laplacian RL refers to representation-learning approaches that build on the low-frequency structure of a Laplacian operator derived from behavior-induced transition data (Wu et al., 2018; Wang et al., 2021). The central idea is to encode long-horizon connectivity: states that are well connected in the data should have nearby embeddings, while states separated by bottlenecks should lie far apart.

The key property behind this is spectral: low-frequency eigencomponents of the Laplacian tend to vary smoothly within each metastable region but exhibit more noticeable changes across bottlenecks. Consequently, the learned embedding provides a geometry that reflects the coarse connectivity structure of the state space, making states within the same region relatively close while helping separate states across bottlenecks.

Formally, in a finite reversible or symmetrized behavior graph, the dataset-induced random walk defines a transition kernel $P$ over states, and the corresponding random-walk Laplacian can be written as $L_{\mathrm{rw}} = I - P$. Its low-frequency eigenvectors $e_i$ capture metastable structure. Mapping a state $s$ to its first $d$ non-trivial components, $\phi(s) = [e_1[s], \ldots, e_d[s]]^\top$, provides an embedding space aligned with the region–bottleneck topology.

In tabular settings one can obtain $\{e_i\}$ by eigendecomposition. In continuous state spaces, however, the Laplacian operator is infinite-dimensional, so we learn $\phi$ by minimizing a spectral graph-drawing objective with orthogonality constraints, estimated from mini-batches of transitions:

$$\min_{\{f_k\}_{k=1}^d} \sum_{k=1}^d \langle f_k, L f_k \rangle \quad \text{s.t.} \quad \langle f_j, f_k \rangle = \delta_{jk}, \ \langle f_k, \mathbf{1} \rangle = 0.$$

Earlier scalable formulations include the unconstrained graph-drawing objective (GDO) (Wu et al., 2018) and the generalized graph-drawing objective (GGDO) that breaks rotational symmetry at the cost of sensitive hyperparameters

(Wang et al., 2021). To avoid these issues, Proper Laplacian Representation Learning introduces the Augmented Lagrangian Laplacian Objective (ALLO) (Gomez et al., 2024), a min–max objective with stop-gradient asymmetry:

$$\max_{\beta} \min_{u \in \mathbb{R}^{d|S|}} \sum_{i=1}^{d} \langle u_i, L\, u_i \rangle$$
$$+ \sum_{j=1}^{d} \sum_{k=1}^{j} \beta_{jk} \big( \langle u_j, [\![ u_k ]\!] \rangle - \delta_{jk} \big)$$
$$+ b \sum_{j=1}^{d} \sum_{k=1}^{j} \big( \langle u_j, [\![ u_k ]\!] \rangle - \delta_{jk} \big)^2.$$

where $\beta_{jk}$ are dual variables, $b > 0$ is a barrier coefficient, and $[\![ \cdot ]\!]$ denotes the stop-gradient operator. This objective is designed to recover ordered eigenvectors and their associated eigenvalues in a stable way. We therefore use its augmented-Lagrangian constraints to enforce orthogonality and stabilize training (details in Appendix).

# 3. Theory in a Nutshell: From Laplacian Spectral Clustering to Optimal Subgoals

**Roadmap and intuition.** We study metastable environments where within-region movement is easy, while progress to distant goals is throttled by a few hard-to-cross bottlenecks. **Result I (bottleneck-guided subgoal optimality):** the next bottleneck is the near-optimal nontrivial one-step subgoal. **Result II (spectral coverage):** when crossing a bottleneck is much harder and rarer than moving inside a region, and the learned Laplacian is accurate enough to reflect this, the low-frequency space provided by Laplacian representation could closely expose the true bottlenecks. Thereby, Laplacian spectral clustering recovers most bottlenecks with small error. **Combining I and II:** thus Laplacian spectral clustering can identify the near-optimal nontrivial one-step subgoal, i.e., the next bottleneck, directly from offline data.

## 3.1. Result I: Bottleneck-guided subgoal optimality

**Theorem 3.1** (Bottleneck-guidance optimality (condensed)). *Given a start $s \in R^{\star}_{\mathrm{cur}}$, a goal set $G \subseteq V \setminus R^{\star}_{\mathrm{cur}}$, and the next mandatory cross-section $\mathcal{B}^{\star}$ on any $s \to G$ path. We restrict the one-step candidate set to admissible progress subgoals*

$$\mathcal{A}_{\mathcal{B}^{\star}} = \{ g \in V : \mathbb{P}_s(\tau_{\mathcal{B}^{\star}} \leq \tau_g) = 1 \},$$

*i.e., candidates that cannot be reached from $s$ without first passing through $\mathcal{B}^{\star}$. Then*

$$\inf_{g \in \mathcal{A}_{\mathcal{B}^{\star}}} \mathcal{J}(g) = T(s \to \mathcal{B}^{\star}) + \mathbb{E}_{\xi}\big[ T(\xi \to G) \big] \pm O\big( t_{\mathrm{mix}} \big),$$

*where $\mathcal{J}(g) := T(s \to g) + T(g \to G)$ and $\xi \sim \mathrm{FirstHit}(s, \mathcal{B}^{\star})$.*

Where $T(x \to A) := \mathbb{E}_x[\tau_A]$ is the expected hitting time, $t_{\mathrm{mix}}$ is the within-region mixing time of the reflected chain on $R^{\star}_{\mathrm{cur}}$, and $\mathrm{FirstHit}(s, \mathcal{B}^{\star})$ is the first-hit distribution on $\mathcal{B}^{\star}$. This restriction excludes no-op or same-region candidates such as $g = s$, which are not meaningful high-level subgoals in our bottleneck-guided planning setting. Proof in Appendix.

**Design implication.** Pick the **next bottleneck** as the one-step subgoal. This is near-optimal whenever moving inside a region is easy and crossing the bottleneck is the main cost.

## 3.2. Result II: Spectral clustering coverage of bottlenecks

**Theorem 3.2** (Spectral clustering coverage of bottlenecks (condensed)). *Given a weighted, undirected graph $\mathcal{G} = (V, W)$ with random-walk kernel $P = D^{-1}W$, Laplacian $L = I - P$, $k$ metastable regions $\{R^{\star}_i\}_{i=1}^k$ whose reflected-chain spectral gaps satisfy $\lambda_2(L^{\mathrm{ref}}_{R^{\star}_i}) \geq \alpha$ and whose inter-region leakage satisfies $\Phi(R^{\star}_i \to R^{\star}_j) \leq \beta \ll \alpha$ for $i \neq j$, eigengap $\gamma = \lambda_{k+1} - \lambda_k > 0$, and an empirical Laplacian $\widehat{L}$ with deviation $\delta = \| \widehat{L} - L \|$. Let $\widehat{\mathcal{R}}$ be obtained by $k$-means on the row-normalized first $k$ eigenvectors of $\widehat{L}$. Then there exist $C_1, C_2, C_3 > 0$ such that*

$$\mathrm{MisVol} \leq C_1 \frac{\beta}{\alpha} + C_2 \frac{\delta}{\gamma},$$
$$\mathrm{Overlap}_{\varepsilon} \geq 1 - C_3 \mathrm{MisVol} - \mu(\mathcal{N}_{\varepsilon}(\partial \mathcal{R}^{\star})).$$

Where $Q(S, T) = \sum_{u \in S, v \in T} \mu(u) P(u, v)$ is inter-set flow, $\Phi(S) = Q(S, S^{\mathrm{c}}) / \mu(S)$ is conductance, $\Phi_{\mathrm{in}}(R)$ is conductance of the reflected chain on $R$, $\mathrm{MisVol} = \min_{\pi \in S_k} \sum_i \mu(\widehat{R}_{\pi(i)} \triangle R^{\star}_i)$ measures mis-clustered volume, $\mathrm{Overlap}_{\varepsilon} = 1 - \mu(\mathcal{N}_{\varepsilon}(\partial \widehat{\mathcal{R}}) \triangle \mathcal{N}_{\varepsilon}(\partial \mathcal{R}^{\star})) / \mu(V)$ measures boundary overlap at tolerance $\varepsilon$, $\mu$ is the stationary distribution of $P$, and $\mathcal{N}_{\varepsilon}(\cdot)$ is an $\varepsilon$-neighborhood in the graph metric. Proof in Appendix.

**Design implication.** Learn a Laplacian embedding and cluster it. When (i) crossing a bottleneck is much rarer/harder than moving within a region, and (ii) the learned embedding faithfully reflects these transition patterns, the resulting cluster boundaries closely match the true bottlenecks.

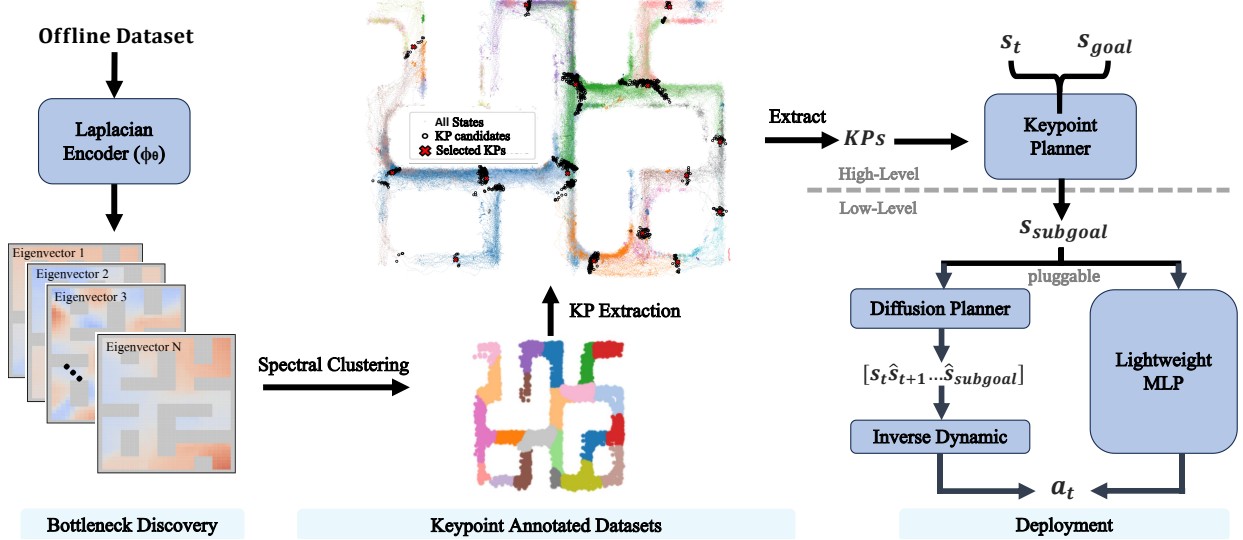

*Figure 2.* BASS overview. We learn a Laplacian encoder $\phi_\theta$ and apply spectral clustering to partition the state space into different regions, whose boundaries expose bottlenecks. We then identify trajectories from the offline dataset that cross these boundaries and use the corresponding boundary-supported states as KP candidates. The most representative KPs are then instantiated from these candidates, and a directed KP reachability graph $\mathcal{G}_{\text{KP}}$ is constructed based on the resulting KPs. Given $(s_t, g)$, a keypoint planner performs KP routing on $\mathcal{G}_{\text{KP}}$, restricts choices to $\mathcal{V}$, and selects the next KP. A pluggable low-level controller, Decision Diffuser or a lightweight MLP, drives the system into the acceptance region $\mathcal{N}(\hat{kp})$, and repeating this over KPs eventually reaches the goal.

---

**Takeaway**

The next bottleneck is the right one-step subgoal, and spectral clustering on a learned Laplacian can recover those bottlenecks under mild, data-driven conditions. Therefore, choosing subgoals at the discovered bottlenecks yields near-optimal plans with a small, interpretable gap.

## 4. Method

In this paper, we propose BASS (**B**ottleneck-**A**ware **S**pectral **S**ubgoaling) for OGCRL in environments where the state space consists of locally connected regions linked by a few hard-to-cross bottlenecks. Since crossing these bottlenecks dominates both time cost and failure risk, BASS follows a simple principle: find bottlenecks, then route through them. As shown in Fig. 2, we reveal bottlenecks from the offline dataset via Laplacian spectral clustering and instantiate a dictionary of *keypoints* (KPs). Formally, we denote the state space by $\mathcal{S} \subseteq \mathbb{R}^D$ and learn a Laplacian encoder $\phi_\theta : \mathcal{S} \to \mathbb{R}^d$, with KPs given by $\hat{kp} \in \mathcal{V} = \{\hat{kp}_1, \ldots, \hat{kp}_M\}$.

At deployment, given $(s_0, g)$, BASS routes over the KP graph, selects the next KP, and invokes a low-level controller to drive the system into the KP's local acceptance region, denoted by $\mathcal{N}(\hat{kp})$ and defined by a simple distance-based criterion.

### 4.1. Discover bottlenecks

From the offline dataset, we construct three artifacts that the deployment stage consumes: a Laplacian encoder $\phi_\theta$, a set of bottleneck keypoints $\mathcal{V}$, and a directed KP reachability graph $\mathcal{G}_{\text{KP}}$. We first learn $\phi_\theta$ using the Proper Laplacian ALLO objective to obtain a low-frequency Laplacian embedding, so that regions become nearly flat and bottlenecks become sharp in the embedding. Applying $K$-Means with a mildly over-segmented number of clusters $K$ to $\phi_\theta(s)$ assigns region labels and reveals boundaries that align with bottlenecks. Using these labels, we then sweep trajectories: whenever a transition $(s_t \to s_{t+1})$ crosses clusters and the new cluster persists for at least $\tau$ steps, we record the local boundary event around $s_{t+1}$ as a KP candidate; representative KPs instantiated from nearby candidates yield the KP set $\mathcal{V}$. Finally, we build the directed graph $\mathcal{G}_{\text{KP}} = (\mathcal{V}, \mathcal{E})$ by connecting $i \to j$ if the dataset contains a short successful fragment such that, starting from the first hit of $\mathcal{N}(\hat{kp}_i)$, the trajectory first hits $\mathcal{N}(\hat{kp}_j)$ without entering any other KP region. Each edge thus encodes a single, data-supported hop between successive bottlenecks. We provide pseudocode for KP discovery in Appendix.

### 4.2. KP semantics and routing

Given $(s_0, g)$ and the offline artifacts $(\phi_\theta, \mathcal{V}, \mathcal{G}_{\text{KP}})$, we compute a KP route $\hat{kp}_{i_0} \to \hat{kp}_{i_1} \to \cdots$ and return the *next* KP for execution. We drop the state coordinates that do not change at the KP and keep only those that do, which makes

the KP easier to reuse on unseen goals. Formally, we represent each KP as

$$\mathrm{KP} = (\mathcal{I}_\Delta, v_\Delta), \qquad \mathcal{I}_\Delta \subseteq \{1, \ldots, D\}, \quad v_\Delta \in \mathbb{R}^{|\mathcal{I}_\Delta|},$$

meaning that the KP specifies desired values $v_\Delta$ on the coordinate slice $\mathcal{I}_\Delta$, while leaving other coordinates unconstrained for routing. The set $\mathcal{I}_\Delta \subseteq \{1, \ldots, D\}$ is automatically selected from the offline dataset, here $D$ denotes the state dimension. Starting from the anchor of $s_0$, we then choose the next KP by running graph search on $\mathcal{G}_{\mathrm{KP}}$ to obtain a route from the current state toward the goal. Starting from the anchor KP closest to the start state and ending at the KP closest to the goal, this yields a compact KP route under the chosen graph-search criterion. We implement this step with lightweight graph search: BFS with simple pruning is used for unit-cost KP graphs (see Appendix), while other simple shortest-path solvers such as Dijkstra can be used in more complex environments when needed.

### 4.3. Pluggable low-level controllers

Once the next KP is selected, the controller only needs to drive the system into its acceptance region. We instantiate two interchangeable choices trained offline and selected by task demands at test time: (i) a Decision Diffuser that predicts a short state rollout $(s_t, \ldots, s_{t+k})$ and is paired with a lightweight inverse-dynamics MLP to recover actions from $(s_t, s_{t+1})$, and (ii) a lightweight MLP that maps $(s_t, \text{next KP})$ directly to $a_t$ for fast inference. Inspired by the fixed-horizon subgoal interface used in HIQL (Park et al., 2023), we optionally train a small MLP regressor to predict an intermediate state $\tilde{s}_{t+k}$, which stabilizes local planning and shortens diffusion horizons.

When the diffusion route is optionally used, short trajectory segments are generated by simulating a reverse-time stochastic differential equation (SDE). Let $\mathbf{x}_t$ denote the vectorized planned trajectory at diffusion time $t$ and $q_t(\mathbf{x}_t)$ the diffused trajectory distribution. The reverse process follows:

$$\mathrm{d}\mathbf{x}_t = \left[ f(t)\,\mathbf{x}_t - g(t)^2 \, \nabla_\mathbf{x} \log q_t(\mathbf{x}_t) \right] \mathrm{d}t + g(t)\,\mathrm{d}\bar{\mathbf{w}}_t,$$

where the score $\nabla_\mathbf{x} \log q_t(\mathbf{x}_t)$ is approximated by the diffusion model's learned denoiser. We condition this process on the current state $s_t$ and an intermediate waypoint $\tilde{s}_{t+k}$ at horizon $k$, then obtain $(s_t, \ldots, s_{t+k})$ and take $a_t = I(s_t, s_{t+1})$ via the inverse-dynamics model.

**Summary.** Offline we learn a Laplacian embedding, expose bottlenecks by clustering, extract KPs, and build $\mathcal{G}_{\mathrm{KP}}$. We route over the KP graph with lightweight graph search, and a pluggable controller executes each hop into the next acceptance region.

## 5. Experiments

### 5.1. Setup

**Environment.** We evaluate on a unified suite of long-horizon, sparse-reward *offline* benchmarks spanning both navigation and manipulation, drawing from widely used D4RL and OGBench tasks. Concretely: **Maze2D/PointMaze/AntMaze/HumanoidMaze** require navigating complex maps (*umaze/medium/large/ultra/giant*) under sparse goal rewards, **FrankaKitchen** requires manipulating the scene by executing any four of seven object-centric skills to reach a target configuration.

Our datasets cover diverse regimes, including *play/diverse*, *stitch* and *partial* (test-time trajectories are longer than training snippets), and *explore* (low-quality data).

**Baselines.** We evaluate on two benchmark suites and align the baselines accordingly. On D4RL, we compare against representative offline methods spanning four paradigms: goal-conditioned imitation **RvS-G** (Emmons et al., 2021), sequence models **Trajectory Transformer (TT)** (Janner et al., 2021), OGCRL methods **HIQL** (Park et al., 2023) and **ESD** (Zhang et al., 2025), and diffusion-based decision making **Diffusion-QL** (Wang et al., 2022), **IDQL** (Hansen-Estruch et al., 2023), **Decision Diffuser (DD)** (Ajay et al., 2022), **Diffuser** (Janner et al., 2022), and **DIAR** (Park et al., 2024a). On OGBench, we report the benchmark's reference OGCRL baselines: goal-conditioned behavioral cloning (**GCBC**) (Lynch et al., 2020; Ghosh et al., 2019), goal-conditioned implicit $V$-learning and $Q$-learning (**GCIVL** and **GCIQL**) (Kostrikov et al., 2021; Park et al., 2023), Quasimetric RL (**QRL**) (Wang et al., 2023), Contrastive RL (**CRL**) (Eysenbach et al., 2022), and **HIQL** (Park et al., 2023).

### 5.2. Main Results

Across a wide variety of navigation and manipulation tasks, scales, and dataset regimes, our method achieves strong and often state-of-the-art performance, especially on bottleneck-dominated long-horizon tasks, highlighting the advantage of using bottlenecks as subgoals. We also try to explain where the gains arise: (i) in precision-sensitive settings (e.g., *AntMaze* corners where high-DoF agents often stumble, *Kitchen* grasps that easily miss), placing subgoals *on* bottlenecks allows the agent to make fine adjustments to enter the KP acceptance region, (ii) in time-limit–prone layouts (*PointMaze/HumanoidMaze* with many detour traps), KP routing finds short or low-cost KP chains, and bottleneck-anchored subgoals steer the agent onto the correct corridor early, avoiding costly backtracking. Beyond success rates, we also compare the evaluation trajectory lengths between BASS and HIQL across multiple environments. Appendix reports that BASS achieves shorter paths on the evaluated

*Table 1.* **Performance comparison on D4RL** (AntMaze/FrankaKitchen: success rate %; Maze2D: D4RL normalized score; higher is better). Baseline numbers are taken from prior reports. For BASS, we report mean±std across 3 seeds. Best in **bold**. "–" indicates not reported by prior work.

| Dataset | TT | RvS-G | HIQL | ESD | Diffusion-QL | IDQL | Diffuser | DD | DIAR | BASS (Ours) |
|---|---|---|---|---|---|---|---|---|---|---|
| *AntMaze (Play/Diverse)* | | | | | | | | | | |
| antmaze-umaze-v2 | **100.0** | 65.4 | 83.3 | 97.1±2.6 | 93.4±3.4 | 94.0 | 0.0 | 0.0 | – | 99.3±0.9 |
| antmaze-umaze-diverse-v2 | – | 60.9 | 85.4 | 92.9±4.2 | 66.2±8.6 | 80.2 | 0.0 | 0.0 | 88.8±1.5 | **98.0±1.6** |
| antmaze-medium-play-v2 | **100.0** | 58.1 | 86.8 | 90.8±6.4 | 76.6±10.8 | 84.5 | 0.0 | 0.0 | – | 98.0±0.0 |
| antmaze-medium-diverse-v2 | 93.3 | 57.3 | 84.1 | 88.3±6.0 | 78.6±10.3 | 84.8 | 0.0 | 0.0 | 68.2±6.7 | **96.7±0.9** |
| antmaze-large-play-v2 | 60.0 | 32.4 | 88.2 | 88.8±6.0 | 46.4±8.3 | 63.5 | 0.0 | 0.0 | – | **96.0±1.6** |
| antmaze-large-diverse-v2 | 66.7 | 36.9 | 86.1 | 87.9±5.0 | 56.6±7.6 | 67.9 | 0.0 | 0.0 | 60.6±2.4 | **98.7±1.9** |
| antmaze-ultra-play-v2 | 33.3 | – | 52.9 | 56.7±9.1 | – | – | 0.0 | 0.0 | – | **97.3±0.9** |
| antmaze-ultra-diverse-v2 | 20.0 | – | 39.2 | 55.8±11.3 | – | – | 0.0 | 0.0 | – | **88.0±1.6** |
| **Average (AntMaze)** | – | – | 75.7 | 82.3 | – | – | 0.0 | 0.0 | – | **96.5** |
| *FrankaKitchen* | | | | | | | | | | |
| kitchen-partial-v0 | – | – | 65.0 | 69.8±2.1 | 60.5±6.9 | – | 56.2 | 57.0 | 63.3±0.9 | **83.3±5.0** |
| kitchen-mixed-v0 | – | – | 67.7 | 67.1±5.0 | 62.6±5.1 | – | 50.0 | 65.0 | 60.8±1.4 | **86.0±2.8** |
| **Average (Kitchen)** | – | – | 66.4 | 68.5 | 61.6 | – | 53.1 | 61.0 | 62.1 | **84.7** |
| *Maze2D* | | | | | | | | | | |
| maze2d-large-v1 | – | – | – | – | – | 90.1 | 123.0 | – | **200.3±3.4** | 189.3±6.2 |

tasks while maintaining competitive or higher success.

### 5.3. Generalization Across Environments

We evaluate generalization along two axes, organized from upper to lower levels in our hierarchy: **(G1)** High-level transfer: swapping keypoint graphs across domains, and **(G2)** Low-level transfer: controller across AntMaze scales. For (G1), we swap keypoint graphs among three datasets with similar state space, including *PointMaze-large-Stitch*, *AntMaze-large-Stitch*, and *AntMaze-large-Explore*, and test them across domains. For (G2), we take a diffuser-based low-level controller trained on AntMaze-Large-Play and transfer it to other AntMaze scales. Results are summarized in Tab. 3 and Tab. 4.

**(G1) High-level transfer: swapping keypoint graphs across domains.** Tab. 3 shows that exchanging the keypoint graph among the three tasks on the same map often preserves reasonably high performance. We emphasize that this is a diagnostic experiment: for each target environment, the in-domain BASS row serves as the reference, and the other rows simply reuse the same policy with a swapped keypoint graph. Our interpretation is that the KP graph captures the topological backbone of the state space, critical corridors and bottlenecks, rather than the shaping of task-specific rewards. As a result, planning on this graph remains valid even when the task or data source differs, yielding stable success rates.

More interestingly, transferring PointMaze KPs to AntMaze leads to higher success than native AntMaze KPs. We hypothesize that PointMaze's simpler point-mass dynamics produce offline data with smoother intra-region transitions and cleaner inter-region boundaries. This makes graph con-

struction and the Laplacian-based representation more faithful to true connectivity and bottlenecks, leading to more accurate bottleneck subgoals. The resulting performance gain is consistent with Theorem 3.1. When reused in AntMaze, the upper level then proposes subgoals that better align with the topological structure of the maze, while the lower level absorbs the actuation complexity of the ant. This suggests a promising direction: learn KPs in simple domains with rich coverage, and transfer them to more complex domains that share a similar state space and transition structure.

**(G2) Low-level transfer: controller across AntMaze scales.** Tab. 4 demonstrates that a diffuser planner trained on AntMaze-Large-Play generalizes strongly to other map scales when paired with each target's own KPs. Although global layouts differ, the controller receives short-horizon subgoals from the upper level and only needs to execute local, easy-to-learn skills including move-to-subgoal and pass-corridor. This decomposition makes the controller largely insensitive to global map differences and encourages robust, reusable primitives. In other words, accurately recovered bottleneck KPs reduce the lower level to a simpler, transferable control task. This observation is consistent with the combined implication of Theorems 3.1 and 3.2.

### 5.4. Visualization

In Fig. 3, colors delineate metastable regions, black dots mark transitions across bottlenecks, and red crosses are the KPs used by the high-level policy. Keypoints concentrate around intersections and corridor transitions where conductance is low and paths often pass through, validating that spectral clustering recovers bottlenecks. Aligning subgoals with these bottlenecks simplifies the task: high-level routing

*Table 2.* **Performance comparison on OGBench** (success rate %; higher is better). Baseline numbers are taken from the OGBench reports. For BASS, we report mean±std across 3 seeds. Best in **bold**.

| Dataset | GCBC | GCIVL | GCIQL | QRL | CRL | HIQL | BASS (Ours) |
|---|---|---|---|---|---|---|---|
| *PointMaze* | | | | | | | |
| pointmaze-large-navigate-v0 | $29 \pm 6$ | $45 \pm 5$ | $34 \pm 3$ | $86 \pm 9$ | $39 \pm 7$ | $58 \pm 5$ | **97.3±0.9** |
| pointmaze-giant-navigate-v0 | $1 \pm 2$ | $0 \pm 0$ | $0 \pm 0$ | $68 \pm 7$ | $27 \pm 10$ | $46 \pm 9$ | **88.0±4.9** |
| pointmaze-teleport-navigate-v0 | $25 \pm 3$ | **45±3** | $24 \pm 7$ | $4 \pm 4$ | $24 \pm 6$ | $18 \pm 4$ | $22.0 \pm 3.3$ |
| pointmaze-large-stitch-v0 | $7 \pm 5$ | $12 \pm 6$ | $31 \pm 2$ | $84 \pm 15$ | $0 \pm 0$ | $13 \pm 6$ | **99.3±0.9** |
| pointmaze-giant-stitch-v0 | $0 \pm 0$ | $0 \pm 0$ | $0 \pm 0$ | $50 \pm 8$ | $0 \pm 0$ | $0 \pm 0$ | **85.3±2.5** |
| pointmaze-teleport-stitch-v0 | $31 \pm 9$ | **44±2** | $25 \pm 3$ | $9 \pm 5$ | $4 \pm 3$ | $34 \pm 4$ | $42.0 \pm 11.4$ |
| *AntMaze (OGBench variants)* | | | | | | | |
| antmaze-large-stitch-v0 | $3 \pm 3$ | $18 \pm 2$ | $7 \pm 2$ | $18 \pm 2$ | $11 \pm 2$ | $67 \pm 5$ | **81.3±5.7** |
| antmaze-giant-stitch-v0 | $0 \pm 0$ | $0 \pm 0$ | $0 \pm 0$ | $0 \pm 0$ | $0 \pm 0$ | $2 \pm 2$ | **71.3±5.7** |
| antmaze-large-explore-v0 | $0 \pm 0$ | $10 \pm 3$ | $0 \pm 0$ | $0 \pm 0$ | $0 \pm 0$ | $4 \pm 5$ | **72.7±0.9** |
| *HumanoidMaze* | | | | | | | |
| humanoidmaze-large-navigate-v0 | $1 \pm 0$ | $2 \pm 1$ | $2 \pm 1$ | $5 \pm 1$ | $24 \pm 4$ | $49 \pm 4$ | **57.3±2.5** |
| humanoidmaze-giant-navigate-v0 | $0 \pm 0$ | $0 \pm 0$ | $0 \pm 0$ | $1 \pm 0$ | $3 \pm 2$ | $12 \pm 4$ | **62.0±7.5** |
| humanoidmaze-large-stitch-v0 | $6 \pm 3$ | $1 \pm 1$ | $0 \pm 0$ | $3 \pm 1$ | $4 \pm 1$ | $28 \pm 3$ | **50.0±1.6** |
| humanoidmaze-giant-stitch-v0 | $0 \pm 0$ | $0 \pm 0$ | $0 \pm 0$ | $0 \pm 0$ | $0 \pm 0$ | $3 \pm 2$ | **55.3±2.5** |
| *Visual Antmaze* | | | | | | | |
| visual-antmaze-large-navigate-v0 | $4 \pm 0$ | $5 \pm 1$ | $4 \pm 1$ | $0 \pm 0$ | **84±1** | $53 \pm 9$ | $65.3 \pm 2.5$ |
| visual-antmaze-large-stitch-v0 | $24 \pm 3$ | $1 \pm 1$ | $0 \pm 0$ | $1 \pm 1$ | $11 \pm 3$ | $28 \pm 2$ | **40.7±4.1** |

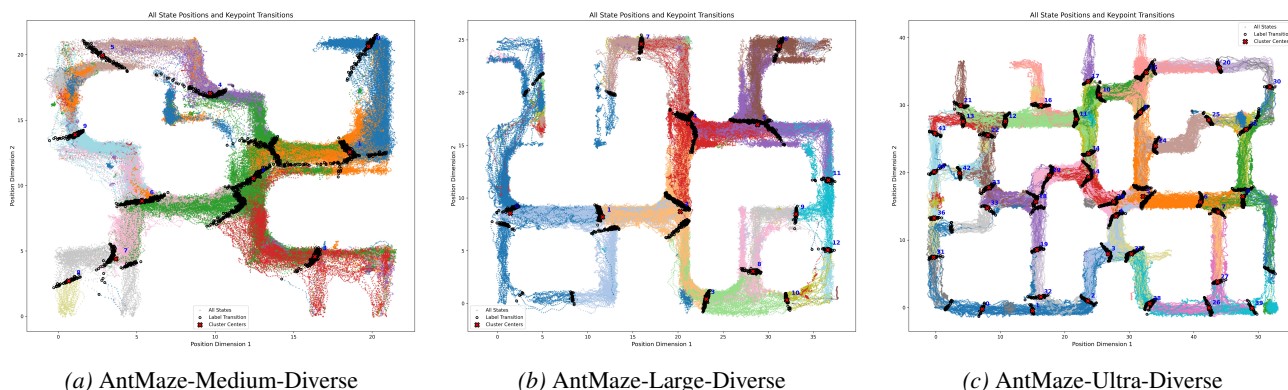

*(a)* AntMaze-Medium-Diverse     *(b)* AntMaze-Large-Diverse     *(c)* AntMaze-Ultra-Diverse

*Figure 3.* Trajectories and keypoints in three AntMaze layouts. Colors indicate metastable regions, black dots denote transition keypoint candidates, red crosses mark selected KPs.

*Table 4.* (G2) Frozen controller transferred across map scales (success %).

| Source map → Target map | Umaze | Medium | Large | Ultra |
|---|---|---|---|---|
| Large | **98.7±0.9** | **96.7±1.9** | **96.0±1.6** | **96.7±0.9** |

*Table 3.* (G1) Cross-domain transfer between PointMaze and AntMaze using swapped keypoint graphs. Rows: KP source; Columns: test environment.

| KP source → Test env | Point-Stitch | Ant-Stitch | Ant-Explore |
|---|---|---|---|
| Point-Stitch | **99.3±0.9** | **83.3±5.2** | **87.3±2.5** |
| AntMaze-Stitch | **99.3±0.9** | $81.3 \pm 5.7$ | $65.3 \pm 4.1$ |
| Ant-Explore | $98.7 \pm 0.9$ | $66.7 \pm 2.5$ | $72.7 \pm 0.9$ |

picks short KP chains, while low-level only needs to enter the next KP's acceptance region. This bottleneck-guided decomposition explains the robust gains observed across scales and datasets. We also hand-annotate oracle keypoints on antmaze-large-diverse at the centers of required corners and visually compare the resulting trajectories with those from BASS, see Appendix.

## 5.5. Ablation Studies

**Ablation of the bottleneck-guided subgoals.** We ablate the high-level subgoal selector. Our method identifies subgoals at bottlenecks via Laplacian spectral clustering. As a drop-in replacement, we use a HIQL-style value-based subgoal selector (Park et al., 2023): at each step, the high level predicts a subgoal associated with a fixed future offset `way_steps`. We test two typical offsets, 25 (HIQL default) and 5.

Tab. 5 shows results on *antmaze-large-play/diverse-v2*. Replacing bottleneck subgoals with the time-based HIQL variant causes substantial drops, especially at the short horizon. This indicates that bottleneck-guided subgoals are the primary driver of our gains. The evidence is also consistent with **Theorem 3.1**, which predicts that bottlenecks are near-optimal subgoals under our assumptions, whereas short-horizon value peaks can be myopic and ignore global connectivity.

*Table 5.* Ablation of the bottleneck-guided subgoals (SGs) on *antmaze-large-play/diverse-v2*

| Setting | Large-Play-v2 | Large-Diverse-v2 |
|---|---|---|
| BASS (ours) | **96.0 ± 1.6** | **98.7 ± 1.9** |
| BASS w/ HIQL-style SG & way_step=25 | 83.3 ± 2.5 | 84.0 ± 4.3 |
| BASS w/ HIQL-style SG & way_step=5 | 18.7 ± 2.5 | 22.7 ± 0.9 |

*Table 6.* The performance of our method with different numbers of clusters on *antmaze-giant-stitch*, *pointmaze-giant-stitch*, *antmaze-large-play* and *pointmaze-large-stitch*.

| clusters | 32 | 36 | 40 | 44 | 48 |
|---|---|---|---|---|---|
| | 52 | 56 | | | |
| antmaze-giant-stitch | 15.3 ± 3.8 | 15.3 ± 3.8 | 52.7 ± 6.6 | 57.3 ± 0.9 | 68.0 ± 5.7 |
| | 62.0 ± 4.3 | 65.3 ± 0.9 | | | |
| pointmaze-giant-stitch | 84.7 ± 3.4 | 87.3 ± 3.4 | 83.3 ± 0.9 | 82.7 ± 1.9 | 84.7 ± 1.9 |
| | 85.3 ± 6.2 | 82.7 ± 0.9 | | | |

| clusters | 10 | 15 | 20 | 24 | 32 |
|---|---|---|---|---|---|
| | 36 | 40 | 44 | 48 | |
| antmaze-large-play | 32.0 ± 3.3 | 15.3 ± 5.7 | 96.0 ± 1.6 | 94.7 ± 2.5 | 95.3 ± 1.9 |
| | 92.0 ± 1.6 | 90.7 ± 3.4 | 94.0 ± 3.3 | 90.0 ± 3.3 | |
| pointmaze-large-stitch | 98.0 ± 1.6 | 100.0 ± 0.0 | 100.0 ± 0.0 | 99.3 ± 0.9 | 100.0 ± 0.0 |
| | 100.0 ± 0.0 | 100.0 ± 0.0 | 100.0 ± 0.0 | 100.0 ± 0.0 | |

**Ablation of the Number of Clusters $K$.** To study how the number of clusters $K$ in Laplacian spectral clustering affects performance, we vary $K$ on four representative environments, results are shown in Table 6. Across these tasks, we observe a consistent pattern: very small $K$ yields overly coarse partitions that under-detect bottlenecks and hurt performance; there is a broad plateau of $K$ where performance is stable and often matches or even exceeds the numbers in the main tables; and only in a few environments does a very large $K$ slightly reduce performance by introducing unnecessary path complexity. This trend is consistent with the intuition behind **Theorem 3.2** that the operative criterion for Laplacian spectral clustering here is to cover bottlenecks.

## 6. Related Work

**Subgoal-Based Goal-Conditioned and Hierarchical RL.** Chane-Sane et al. (2021) introduce imagined subgoals predicted by a high-level policy, corresponding to intermediate states that are approximately halfway to the goal under a learned value/reachability metric. These subgoals regularize policy learning but are not used as explicit test-time waypoints. Another approach (Zhang et al., 2020) constrains subgoals within a $k$-step neighborhood to maintain local feasibility within a limited horizon. In addition, graph-based planning methods also support GCRL. For instance, Kim et al. (2023) use graph-based planning to produce subgoal sequences and distill the resulting subgoal-conditioned behavior into goal-conditioned policies. This combines goal-conditioned policies with graph-based reasoning to facilitate task completion. Meanwhile, diffusion-based methods can generate goal- or constraint-conditioned trajectories for decision making and can be plugged into these frameworks (Ajay et al., 2022; Janner et al., 2022). In addition, hierarchical planning approaches explore subgoal generation using graphs or models. Fang et al. (2023) plan sequences of subgoals in a learned lossy representation space with an affordance model, while Li et al. (2022) combine an offline goal-conditioned low-level policy with a model-based high-level planner and CVAE-sampled subgoal candidates. $CE^2$ (Duan et al., 2024) leverages cluster boundaries in a learned latent space for online goal-directed exploration, while our focus is on offline planning and subgoal selection.

Quasimetric RL (QRL) (Wang et al., 2023) learns quasimetric value/distance representations for goal reaching. HILP (Park et al., 2024b) learns Hilbert representations that preserve temporal structure and enable zero-shot policy prompting. Graph-Assisted Stitching (GAS) (Baek et al., 2025) formulates subgoal selection as graph search in a temporal-distance representation, emphasizing micro-level trajectory stitching across offline data. In contrast, BASS discovers macro-level bottleneck keypoints via Laplacian structure.

**Laplacian Representation.** In Laplacian representation learning for RL, early work Mahadevan & Maggioni (2007) introduced Proto-Value Functions (PVF), leveraging random-walk Laplacian eigenvectors to construct a geometry-aware state representation that captures the large-scale connectivity of the underlying Markov chain. Wu et al. (2018) expanded this by proposing a Graph Drawing Objective (GDO) for large state spaces, but it struggled with

eigenvector rotations and hyperparameter tuning. Wang et al. (2021) introduced the Generalized Graph Drawing Objective (GGDO), improving eigenvector recovery by breaking rotational symmetry, while later work notes remaining sensitivity and eigenvalue-recovery issues in graph-drawing formulations. Gomez et al. (2024) introduced the Augmented Lagrangian Laplacian Objective (ALLO), which addresses these issues by recovering eigenvectors and associated eigenvalues with reduced hyperparameter dependence. In addition, Klissarov & Machado (2023) used Laplacian representations to improve exploration. By contrast, our work uses Laplacian structure to build a bottleneck keypoint graph for long-horizon offline goal-conditioned decision-making, focusing on discovering semantic bottlenecks and routing through them rather than on exploration per se.

## 7. Conclusions

We reframed offline goal-conditioned RL as routing through metastable regions connected by a few hard-to-cross bottlenecks. Our principle is simple: the near-optimal one-step subgoal is the next bottleneck. We operationalize this by learning a Laplacian representation from offline data, applying spectral clustering to expose bottlenecks, extracting keypoints (KPs) from boundary-supported candidates, and planning with a lightweight, dynamics-agnostic graph search over the KP graph. A pluggable low-level controller, either a Decision Diffuser or a lightweight MLP, then drives the system into each KP's acceptance region.

Theory establishes subgoal optimality (Theorem 3.1) and boundary recovery (Theorem 3.2), implying near-optimal routing. Experiments on D4RL and OGBench show strong, often state-of-the-art success and generalize across controllers, domains, and scales, including KP-graph swapping (G1) and controller transfer across AntMaze scales (G2).

## Acknowledgement

This work is supported by the National Natural Science Foundation of China (Grant Nos. 62406271, 62533021, 62422605, 92370132), the National Key Research and Development Program of China (Grant No. 2024YFE0210900), and the Fundamental Research Program of Shanxi Province (Serial No. 202503021212091).

## Impact Statement

This paper presents a study on offline goal-conditioned reinforcement learning. While our work may have broader societal implications through potential downstream applications, we do not identify any specific impacts that require additional discussion here.

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

# A. Pseudo-code for Bottleneck Discovery

---

**Algorithm 1** Bottleneck keypoint discovery

---

1: **Input:** Offline dataset $\mathcal{D}_{\text{off}} = \{\mathbf{h}_i\}$, number of clusters $K$, boundary persistence $\tau$
2: **Output:** Keypoint set KPs
3: **// Laplacian representation and spectral clustering**
4: Train a Laplacian encoder $\phi$ on states $\{s \mid (s, \cdot) \in \mathcal{D}_{\text{off}}\}$ and obtain embeddings $z = \phi(s)$.
5: Run $K$-means with $K$ clusters on $\{z\}$; let $c(s) \in \{1, \dots, K\}$ be the cluster label of state $s$.
6: **// Collect KP candidates**
7: Initialize boundary buffer $\mathcal{B} \leftarrow \emptyset$.
8: **for** each trajectory $h = (s_0, \dots, s_T)$ in $\mathcal{D}_{\text{off}}$ **do**
9:     **for** $t = 0 \dots T - \tau - 1$ **do**
10:         **if** $c(s_t) \neq c(s_{t+1})$ **and** $c(s_{t+1}) = \cdots = c(s_{t+\tau})$ **then**
11:             Append the local event around $s_{t+1}$ to $\mathcal{B}$
12:         **end if**
13:     **end for**
14: **end for**
15: **// Form representative keypoints**
16: Initialize keypoint set KPs $\leftarrow \emptyset$.
17: Organize candidates in $\mathcal{B}$ into small neighborhoods and compute a representative $\mu_\ell$ for each group.
18: **for** each representative $\mu_\ell$ **do**
19:     Construct a keypoint $\text{KP}_\ell = (I_\Delta, v_\Delta)$ from the representative $\mu_\ell$ as described in Sec. 4.2.
20:     Add $\text{KP}_\ell$ to KPs
21: **end for**
22: **Return** KPs

---

# B. Average Evaluation Steps

*Table 7.* Average evaluation steps for HIQL and our method across different environments.

| Dataset | HIQL Steps | BASS (ours) Steps |
|---|---|---|
| **antmaze-giant-stitch-v0** | 997.50 | 864.17 |
| **antmaze-large-stitch-v0** | 640.87 | 547.28 |
| **pointmaze-large-stitch-v0** | 905.04 | 265.33 |
| **humanoidmaze-large-navigate-v0** | 1667.65 | 1652.34 |
| **humanoidmaze-large-stitch-v0** | 1808.40 | 1717.22 |
| **humanoidmaze-giant-navigate-v0** | 3900.32 | 3287.94 |
| **humanoidmaze-giant-stitch-v0** | 3978.45 | 3319.76 |

# C. Trajectory visualization and comparison with expert hand-annotated trajectories

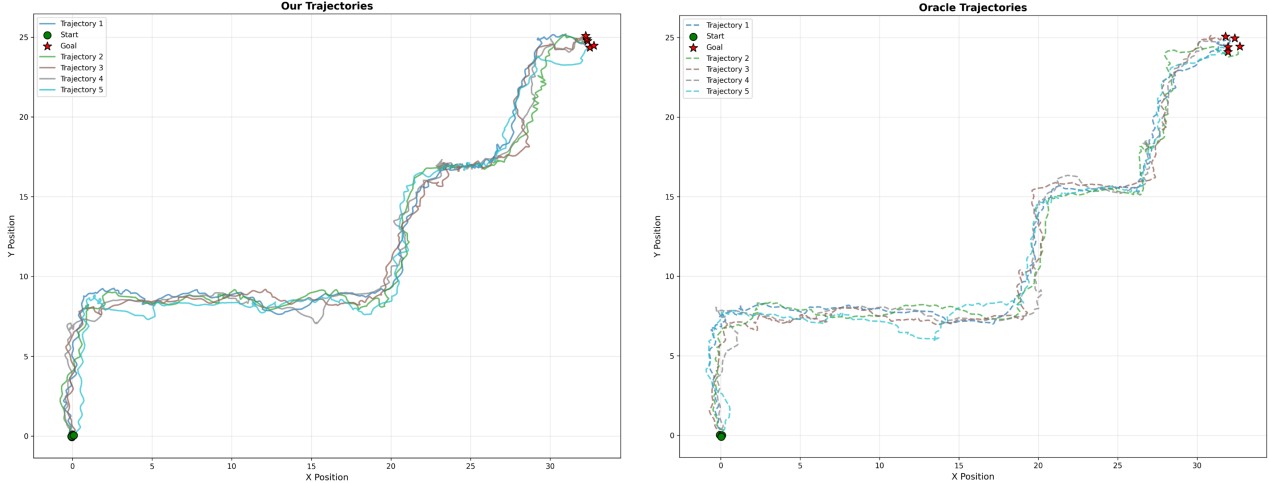

*Figure 4.* Comparison between our trajectories and oracle trajectories.

## D. Implementation Details of the Laplacian Loss

In our framework, the Laplacian representation is learned by minimizing a loss function that creates a feature space reflecting the temporal connectivity of the state space. In this representation space, states that require many transitions to connect (i.e., have long transition durations) are far apart, while states that are easily reachable (i.e., with short transition periods) are embedded close together. Such a design not only naturally measures transition difficulty but also highlights bottlenecks and regions where rapid changes in the learned representation indicate potential sub-task boundaries. These boundaries manifest as clustering limits where keypoints are more likely to occur.

Our implementation follows the ALLO objective and optimization recipe of (Gomez et al., 2024). Training proceeds through four high-level stages:

### D.1. Data Sampling

- **Graph-Drawing (Primal) Pairs:** From the replay buffer or trajectory dataset, we sample temporal pairs $(s_t, s_{t+k})$ from the same trajectory, where $k \geq 1$ is drawn from a future-state sampling distribution.

- **Orthogonality (Constraint) Batches:** Independently sample two batches of states $\{s_i^1\}$ and $\{s_i^2\}$. These batches are not temporally paired; they are used to estimate the lower-triangular orthogonality errors of the embedding dimensions.

### D.2. Representation Encoding

A single encoder network $\phi_\theta$ maps each sampled state into a $d$-dimensional embedding:

$$u = \phi_\theta(s) \in \mathbb{R}^d. \tag{1}$$

- $\phi_\theta(s_t)$ and $\phi_\theta(s_{t+k})$ are used to compute the graph-drawing loss, matching the $\langle u, Lu \rangle$ term of the proper Laplacian.

- $\phi_\theta(s_i^1)$ and $\phi_\theta(s_i^2)$ are used to compute the orthogonality error matrix, implementing the $u^T u = I$ constraint softly.

### D.3. Loss Construction

We combine three terms into a single augmented Lagrangian that exactly mirrors the proper Laplacian objective:

$$\mathcal{L}_{\text{total}} = \mathcal{L}_{\text{graph}} + \mathcal{L}_{\text{dual}} + \mathcal{L}_{\text{barrier}}. \tag{2}$$

- **Graph-Drawing Term:**

$$\mathcal{L}_{\text{graph}} = \sum_{i=1}^{d} \left( u_t^i - u_{t+k}^i \right)^2, \tag{3}$$

  which gives a sampled graph-drawing penalty corresponding to the Laplacian smoothness term.

- **Linear Lagrangian (Dual) Term:** Let

$$C_{jk} = \frac{1}{B} \sum_i \phi_j(s_i) \, \text{stopgrad}(\phi_k(s_i)) - \delta_{jk}, \qquad j \geq k, \tag{4}$$

  denote the sampled lower-triangular orthogonality error. The dual term is

$$\mathcal{L}_{\text{dual}} = \sum_{j \geq k} \beta_{jk} C_{jk}, \tag{5}$$

  with dual variables $\beta_{jk}$ enforcing the orthogonality constraints in the augmented-Lagrangian sense.

- **Barrier Penalty:** Using two independently sampled constraint batches to estimate the quadratic error, we define

$$\mathcal{L}_{\text{barrier}} = b \sum_{j \geq k} C_{jk}^{(1)} C_{jk}^{(2)}, \tag{6}$$

  which softly penalizes violations of the orthogonality constraint.

## D.4. Joint Optimization with Alternating Updates

1. **Encoder Update:** With $\beta$ and $b$ fixed, minimize $\mathcal{L}_{\text{total}}$ w.r.t. $\theta$, exactly following the proper spectral embedding procedure.

2. **Dual Variables Update:** With $\theta$ fixed, perform a projected gradient *ascent* step on $\beta$ using current orthogonality errors, corresponding to the update of Lagrange multipliers.

3. **Barrier Scheduling:** Increase $b$ over training—on a schedule or when constraint violations persist—to maintain the strength of the barrier term, as in augmented Lagrangian methods.

## D.5. Summary

By strictly following the classical Laplacian spectral graph objective and its augmented Lagrangian relaxation—combining

1. a graph-drawing term preserving *transition difficulty*,

2. a linear Lagrangian term enforcing orthonormality,

3. a quadratic barrier penalty for soft constraints,

and by alternating minimization for the encoder with maximization for the duals, we obtain a proper Laplacian embedding that faithfully preserves temporal connectivity and yields disentangled, stable representations for downstream keypoint detection and hierarchical control.

# E. Residual-state BFS for KP routing

This section details the vector-state, unit-cost case, where BFS with simple pruning is natural. Other KP graph-search variants, such as Dijkstra with simple edge costs, are straightforward and use the same KP graph. In the main text (§4.2), we represent a keypoint (KP) as

$$\text{KP} = (\mathcal{I}_\Delta, v_\Delta), \qquad \mathcal{I}_\Delta \subseteq \{1, \dots, D\}, \quad v_\Delta \in \mathbb{R}^{|\mathcal{I}_\Delta|}, \tag{7}$$

which specifies desired values on the coordinates in $\mathcal{I}_\Delta$ while leaving the remaining coordinates unconstrained for routing. Given a start–goal pair $(s_0, g)$, we plan only over coordinates that differ from the goal:

$$\mathcal{R}_0 = \{\, i :\ s_0[i] \neq g[i] \,\}, \qquad q = |\mathcal{R}_0|. \tag{8}$$

For each KP, we keep only its goal-aligned footprint

$$F(\mathrm{KP}; g) = \{\, i \in \mathcal{I}_\Delta :\ v_\Delta[i] = g[i]\}. \tag{9}$$

**Routing.** We perform BFS for the vector-state, unit-cost routing case, maintaining the current residual coordinate set $\mathcal{R}$. When applicable, we use simple pruning heuristics; for example, in some vector-state environments, we may skip KPs whose specified coordinates do not change any currently relevant residual dimensions, i.e., those with $F(\mathrm{KP}; g) \cap \mathcal{R} = \varnothing$.

## F. Low-Level Strategy (Pluggable)

Our low level is *modular* and exposes a unified interface

$$a_t = \texttt{LOWLEVEL}(s_t,\ \tilde{g}_t;\ \eta), \tag{10}$$

where $\tilde{g}_t$ is the high-level keypoint–guided mid-goal and $\eta$ are backend hyperparameters. We run in a receding horizon: compute $a_t$ from $(s_t, \tilde{g}_t)$, step the env to get $s_{t+1}$, and repeat.

### F.1. Keypoint-Conditioned $k$-Step State Prediction

Because time-to-reach a keypoint is uncertain while planners often assume a fixed horizon $k$, in environments where this planner is used, we first predict a *$k$-step target state* $s_{t+k}$ conditioned on the current state and the selected keypoint:

$$s_{t+k} = \pi_\omega\big(s_t,\ k_i\big), \tag{11}$$

where $k_i$ is the keypoint selected by the high level. Concretely:

- **Inputs.** $(s_t, k_i)$.

- **Objective.** A value model $V_\phi(s, k)$ provides HIQL-style supervision to train $\pi_\omega$ so that the predicted $s_{t+k}$ maximizes the expected keypoint-conditioned return over $k$ steps.

- **Output.** $s_{t+k}$, which anchors a short-horizon plan.

This $k$-step target is then consumed by one of two interchangeable backends.

### F.2. Backend A: Short-Horizon Diffusion Planner (Decision Diffuser)

**Conditioning.** Generate a $k$-step local plan from $s_t$ to the target $s_{t+k}$ by conditional diffusion, using $(s_t, s_{t+k})$ (or $(s_t, \tilde{g}_t)$ if planning in action space) as conditioning signals.

**Sampling.** A time-indexed network $\epsilon_\theta(\cdot, t)$ approximates the reverse score to produce a smooth trajectory $\{s_t, \ldots, s_{t+k}\}$ with a small number of reverse steps.

**State→Action.** If planning in *state space*, actions are recovered via a lightweight inverse-dynamics MLP $I_\zeta$:

$$a_t = I_\zeta\big(s_t, s_{t+1}\big), \tag{12}$$

trained with MSE on offline transitions. If planning directly in *action space*, $I_\zeta$ is not used.

### F.3. Backend B: Goal-Conditioned Reactive Controller (GC-MLP)

**Inputs.** Concatenate the current state and subgoal: $x_t = [s_t, \tilde{g}_t]$ (or $[s_t, s_{t+k}]$). A small MLP $f_\psi$ outputs $a_t = f_\psi(x_t)$.

**Training: IQL Objective.** We train the GC-MLP with the IQL loss, together with a value network $V_\phi$ and a critic $Q_\theta$. For notational simplicity, we omit the waypoint argument in $V_\phi$ and $Q_\theta$ below.

$$\mathcal{L}_Q(\theta) = \mathbb{E}_{(s,a,r,s')\sim\mathcal{D}}\Big[\big(Q_\theta(s,a) - \big(r + \gamma\, V_\phi(s')\big)\big)^2\Big], \tag{13}$$

$$\mathcal{L}_V(\phi) = \mathbb{E}_{(s,a)\sim\mathcal{D}}\Big[\rho_\tau\big(Q_\theta(s,a) - V_\phi(s)\big)\Big], \quad \rho_\tau(\delta) = |\tau - \mathbb{1}\{\delta < 0\}|\,\delta^2, \tag{14}$$

$$\mathcal{L}_\pi(\psi) = \mathbb{E}_{(s,a)\sim\mathcal{D}}\Big[\exp\big(\tfrac{Q_\theta(s,a) - V_\phi(s)}{\beta}\big)\,\|a - f_\psi([s,\tilde{g}])\|_2^2\Big], \tag{15}$$

where $\tau$ is the expectile level and $\beta$ is the temperature. At test time we condition $f_\psi$ on either $\tilde{g}_t$ or $s_{t+k}$ depending on the configuration.

### F.4. Summary

- **Low-level control pipeline.** When the fixed-horizon planner is used, we first predict a keypoint-conditioned $k$-step target $s_{t+k}$ and then realize control with a diffusion planner, optionally followed by inverse dynamics. In environments using the lightweight controller, we instead use a GC-MLP trained with the IQL objective.

- **Pluggability.** Both backends implement the same interface $a_t = \texttt{LOWLEVEL}(s_t, \tilde{g}_t; \eta)$ and can be swapped without changing the high level.

- **Effect.** Bottleneck-guided subgoals provide reliable waypoints, so the low level only needs to execute short, simple transitions between keypoints.

## G. Hyperparameters

We summarize the hyperparameters in Tab. 8. In all experiments we follow the ALLO configuration of (Gomez et al., 2024) for the Laplacian encoder, except that we increase the non-trivial embedding dimension from 10 to 20 for most environments. Unless otherwise specified, BASS results in the experimental tables are reported as mean $\pm$ standard deviation over three seeds, with the standard deviation computed using `ddof=0`.

*Table 8.* Hyperparameters

| Hyperparameter | Value & Specifics |
| --- | --- |
| d (embedding dim.) | 20 |
| other Laplacian representation params | follow (Gomez et al., 2024) settings |
| k (intermediate horizon) | 5 for AntMaze(d4rl), 25 for Kitchen |
| T (diffusion steps) | 5 for AntMaze(d4rl), 50 for Kitchen |
| Diffusion model | DiT with hidden_dim=384, nhead=8, layers=3 |
| Optimizer (diffusion) | AdamW with lr=$2 \times 10^{-4}$ |
| Optimizer (ivdm) | weight_decay=$1 \times 10^{-5}$ |
| Inverse dynamics (hidden_size) | MLP hidden_size=256, optimizer Adam lr=$2 \times 10^{-4}$ |

## H. Laplacian Representation

In this section, we present a series of visualizations of the Laplacian representation in various antmaze environments. The figures illustrate both the learned eigenvectors and the results of spectral clustering.

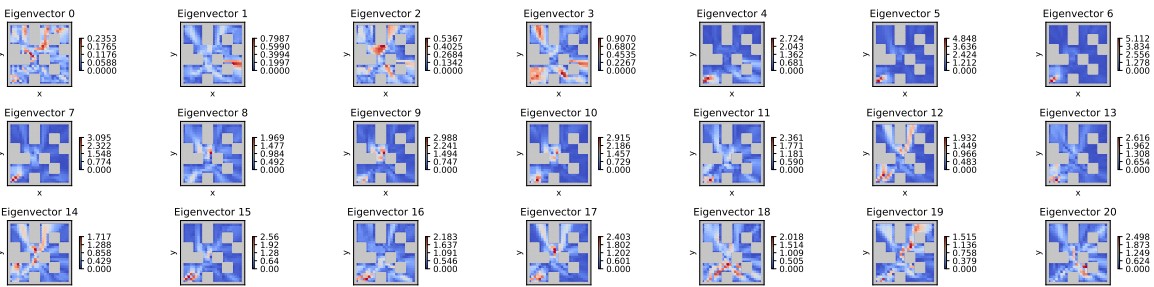

*Figure 5.* Learned eigenvector gradients for antmaze-medium-play

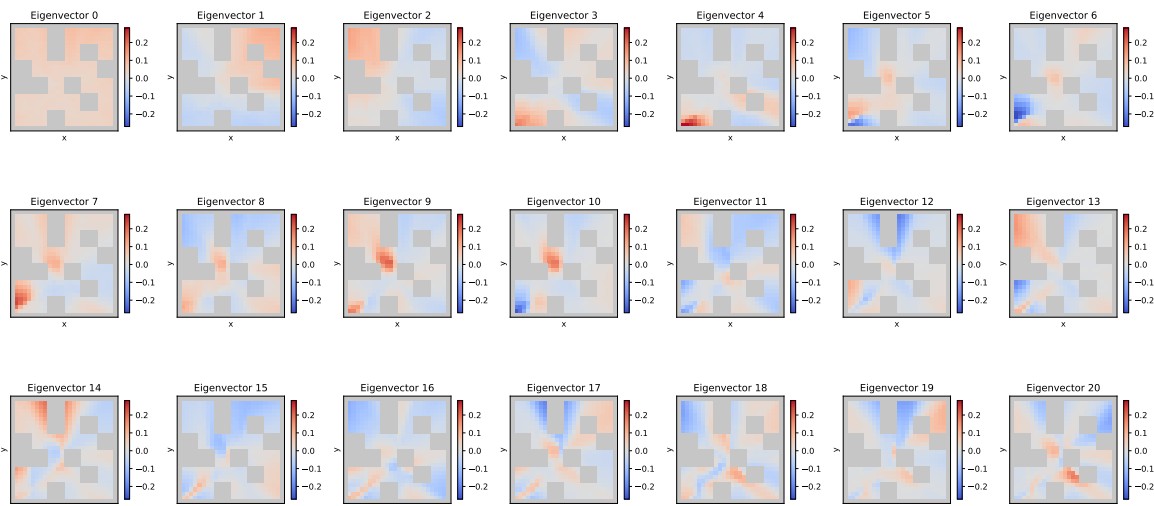

*Figure 6.* Learned eigenvectors for antmaze-medium-play

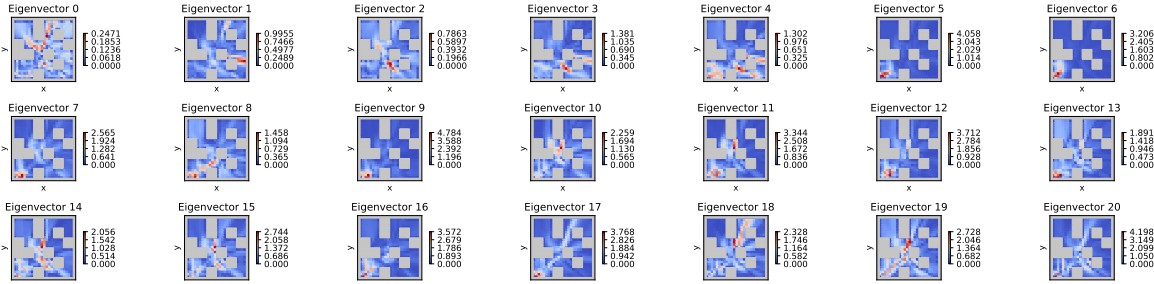

*Figure 7.* Learned eigenvector gradients for antmaze-medium-diverse

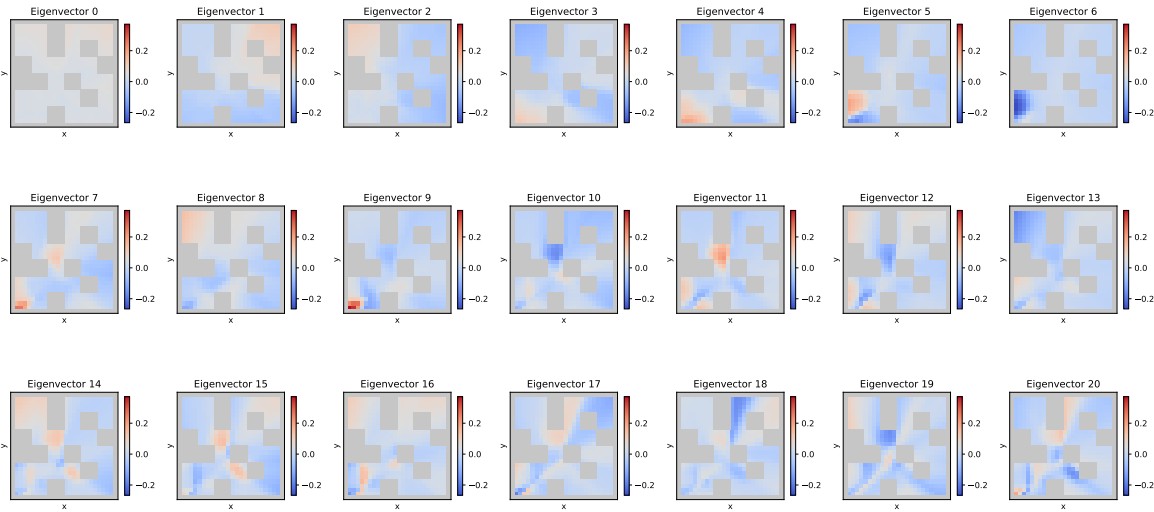

*Figure 8.* Learned eigenvectors for antmaze-medium-diverse

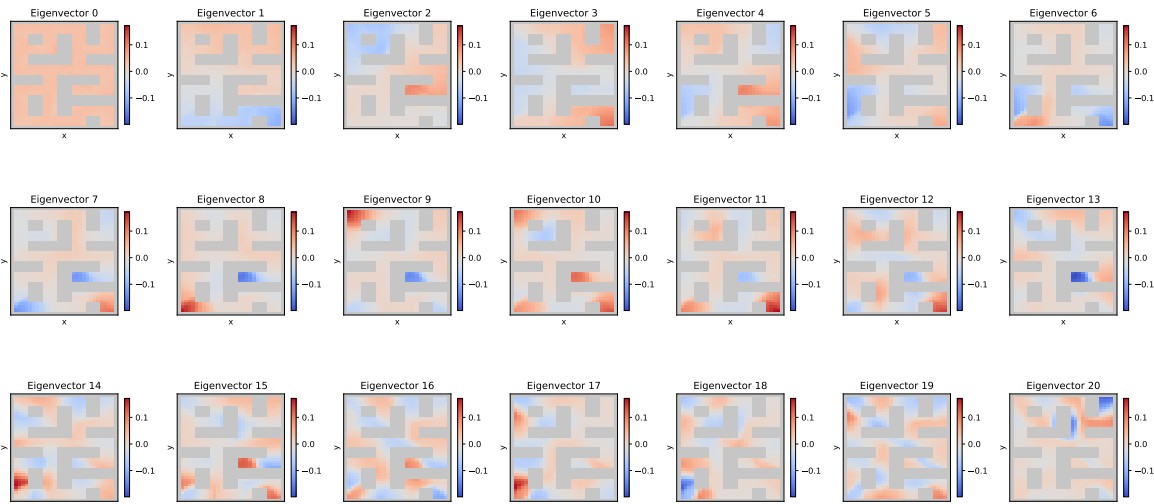

*Figure 9.* Learned eigenvectors for antmaze-large-play

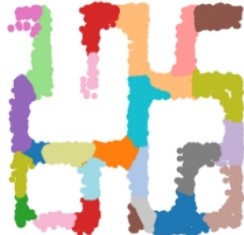

*Figure 10.* Spectral clustering results for antmaze-large-play

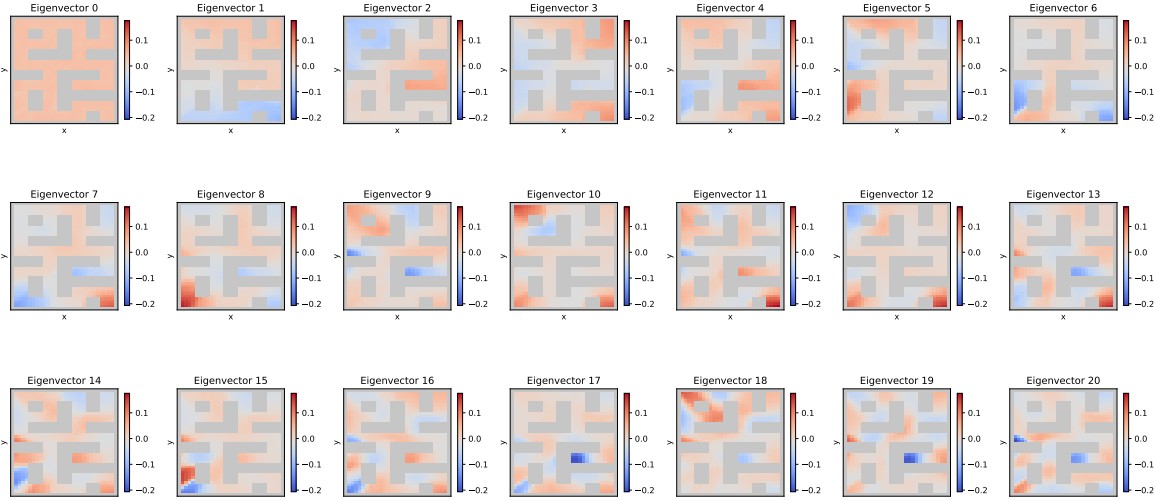

*Figure 11.* Learned eigenvectors for antmaze-large-diverse

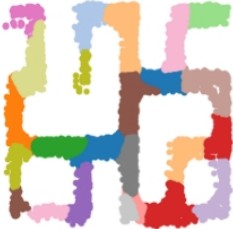

*Figure 12.* Spectral clustering results for antmaze-large-diverse

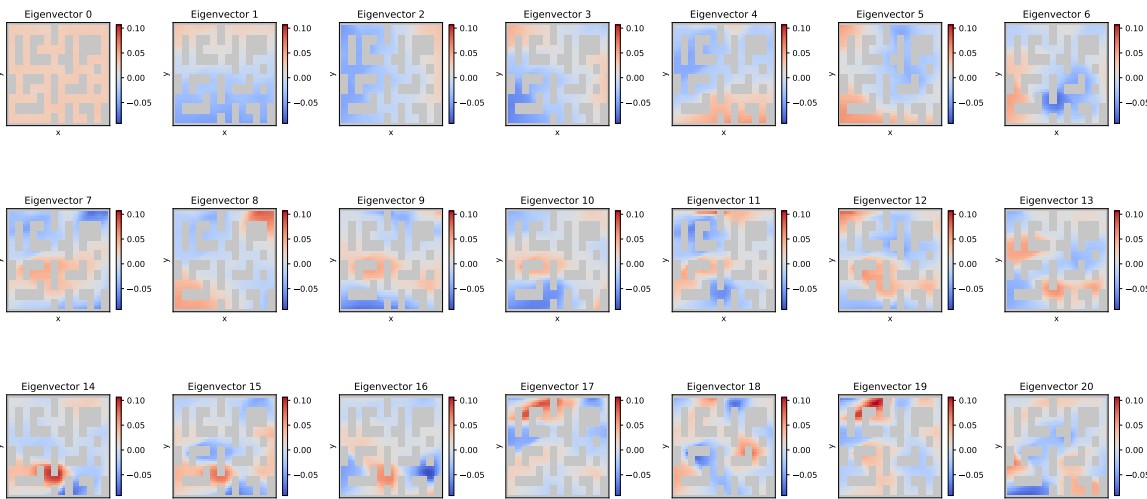

*Figure 13.* Learned eigenvectors for antmaze-ultra-play

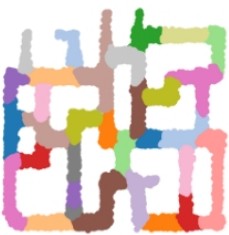

*Figure 14.* Spectral clustering results for antmaze-ultra-play

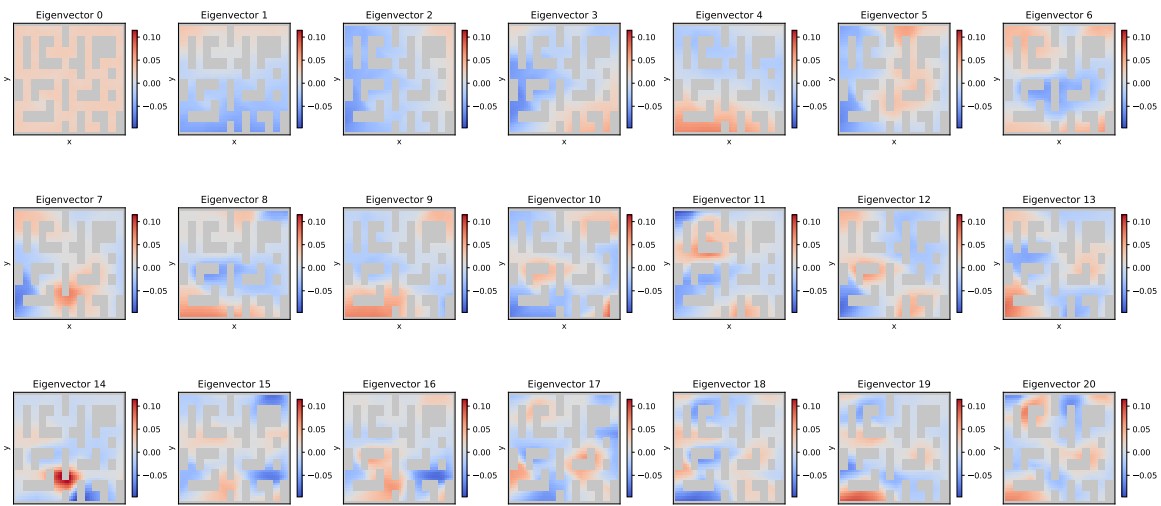

*Figure 15.* Learned eigenvectors for antmaze-ultra-diverse

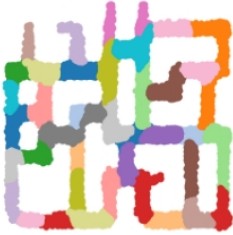

*Figure 16.* Spectral clustering results for antmaze-ultra-diverse

# I. Proofs and Technical Details for Theories

This appendix expands the statements in Section 3, provides self-contained proofs under standard assumptions, and aligns the notation with the main text. Throughout we work on a weighted, undirected state-transition graph $G = (V, W)$ built from offline data, with degree $D = \operatorname{diag}(W\mathbf{1})$, random-walk kernel $P = D^{-1}W$, and random-walk Laplacian $L = I - P$. The (unique) stationary distribution is $\mu^\top P = \mu^\top$, $\sum_{v \in V} \mu(v) = 1$. Eigenvalues of $L$ satisfy $0 = \lambda_1 \le \lambda_2 \le \cdots$, and the $k$-way eigengap is $\gamma = \lambda_{k+1} - \lambda_k > 0$. For measurable $S, T \subseteq V$ define the inter-set flow

$$Q(S, T) = \sum_{u \in S, v \in T} \mu(u)P(u, v), \tag{16}$$

the (outer) conductance $\Phi(S) = Q(S, S^c)/\mu(S)$, and, for a region $R$, the internal conductance of the *reflected* chain $P_R$ (with stationary law $\mu_R$),

$$\Phi_{\mathrm{in}}(R) = \inf_{\emptyset \ne A \subseteq R,\ \mu_R(A) \le \frac{1}{2}} \frac{Q_R(A, R\backslash A)}{\mu_R(A)}. \tag{17}$$

For $A \subseteq V$, write $\partial A = \{v \in V : P(v, A) > 0 \text{ and } P(v, A^c) > 0\}$ and $\overline{A} = A \cup \partial A$. For any $B \subseteq V$ and $\varepsilon > 0$, $\mathcal{N}_\varepsilon(B)$ denotes the $\varepsilon$-neighborhood in the shortest-path (graph) metric. We use the following metastability condition (inner-strong / outer-weak):

$$\exists\, \mathcal{R}^\star = \{R_1^\star, \ldots, R_k^\star\} \text{ with } \mu(R_i^\star) \in [\eta, 1-\eta], \quad \Phi_{\mathrm{in}}(R_i^\star) \ge \alpha, \quad \Phi(R_i^\star \to R_j^\star) \le \beta \ll \alpha\ (i \ne j). \tag{18}$$

Let $\widehat{L}$ be the Laplacian estimated from offline data and $\delta = \|\widehat{L} - L\|$ its operator-norm deviation.

**Hitting times and mixing.** For $A \subseteq V$, let $\tau_A = \inf\{t \ge 0 : X_t \in A\}$ be the hitting time, and define $T(x \to A) = \mathbb{E}_x[\tau_A]$. For a region $R$, let $t_{\mathrm{mix}}(R)$ be the least $t$ such that $\max_{x \in R} \|P_R^t(x, \cdot) - \mu_R\|_{\mathrm{TV}} \le 1/4$; we write $t_{\mathrm{mix}}$ for $t_{\mathrm{mix}}(R_{\mathrm{cur}}^\star)$ when context is clear.

## I.1. Bottleneck-guided subgoal optimality (full version of Thm. 3.1)

**Theorem I.1** (Bottleneck-first optimality). *Fix a start $s \in R_{\mathrm{cur}}^\star$ and a goal set $G \subseteq V \setminus R_{\mathrm{cur}}^\star$. Consider the one-step high-level objective over admissible progress candidates*

$$J(g) := T(s \to g) + T(g \to G), \quad g \in A_{B^\star}.$$

*where $A_{B^\star} := \{g \in V : \mathbb{P}_s(\tau_{B^\star} \le \tau_g) = 1\}$. Let $\mathbb{B}^\star$ denote a next mandatory cross-section for any $s \to G$ path (e.g., an $s$–$G$ minimum-capacity cut intersected with $\partial R_{\mathrm{cur}}^\star$), and let $\xi \sim \mathrm{FirstHit}(s, \mathbb{B}^\star)$ be the first-hit distribution on $\mathbb{B}^\star$. Assume that the bottleneck cross-section has $O(t_{\mathrm{mix}})$ internal hitting diameter. Then there exists $g^\star \in B^\star$ such that*

$$\inf_{g \in A_{B^\star}} J(g) = T(s \to B^\star) + \mathbb{E}_\xi[T(\xi \to G)] \pm C \cdot t_{\mathrm{mix}},$$

*where $C > 0$ depends on the chosen total-variation threshold and the local bottleneck-diameter constant.*

*Proof sketch.* (*Decomposition at the bottleneck*) For any admissible progress candidate $g \in \mathcal{A}_{B^\star}$, we have $\mathbb{P}_s(\tau_{\mathbb{B}^\star} \le \tau_g) = 1$. Therefore, by the strong Markov property at $\tau_{\mathbb{B}^\star}$,

$$T(s \to g) = T(s \to \mathbb{B}^\star) + \mathbb{E}_\xi[T(\xi \to g)], \tag{19}$$

hence

$$\mathcal{J}(g) = T(s \to \mathbb{B}^\star) + \mathbb{E}_\xi[T(\xi \to g) + T(g \to G)]. \tag{20}$$

(*Lower bound*) By the triangle inequality for hitting times, $T(\xi \to g) + T(g \to G) \ge T(\xi \to G)$ for any $g \in \mathcal{A}_{B^\star}$, yielding

$$\mathcal{J}(g) \ge T(s \to \mathbb{B}^\star) + \mathbb{E}_\xi[T(\xi \to G)]. \tag{21}$$

(*Achievability up to local bottleneck diameter*) Pick $g \in B^\star$. By the assumed $O(t_{\mathrm{mix}})$ internal hitting diameter of $B^\star$, the distance from the first-hit point $\xi$ to $g$ is controlled by $O(t_{\mathrm{mix}})$. Moreover, for any $x \in B^\star$, the triangle inequality for hitting times gives

$$T(g \to G) \le T(g \to x) + T(x \to G),$$

and therefore $T(g \to G) \le \mathbb{E}_\xi[T(\xi \to G)] + O(t_{\mathrm{mix}})$. Combining with the decomposition gives the claim. $\qquad \square$

**Design implication (restated).** Placing the *next bottleneck* as the one-step subgoal is near-optimal up to an $O(t_{\mathrm{mix}})$ gap whenever movement inside a region is fast compared with crossing the bottleneck.

## I.2. Spectral clustering coverage of bottlenecks (full version of Thm. 3.2)

Let $U \in \mathbb{R}^{|V| \times k}$ collect the first $k$ eigenvectors of $L$, and $Z$ be its row-normalization (each row scaled to unit $\ell_2$ norm). Likewise obtain $\widehat{U}, \widehat{Z}$ from $\widehat{L}$. Running $k$-means on the rows of $\widehat{Z}$ returns a partition $\widehat{\mathcal{R}} = \{\widehat{R}_1, \ldots, \widehat{R}_k\}$. Define the *misclustered volume* (up to permutation $\pi$) and the $\varepsilon$-thick *bottleneck overlap*:

$$\mathrm{MisVol} = \min_{\pi \in S_k} \sum_{i=1}^{k} \mu\big(\widehat{R}_{\pi(i)} \triangle R_i^\star\big), \qquad \mathrm{Overlap}_\varepsilon = 1 - \frac{\mu\big(\mathcal{N}_\varepsilon(\partial\widehat{\mathcal{R}}) \triangle \mathcal{N}_\varepsilon(\partial\mathcal{R}^\star)\big)}{\mu(V)}. \tag{22}$$

**Theorem I.2** (High-overlap bottleneck recovery). *Under metastability equation 18 with eigengap $\gamma = \lambda_{k+1} - \lambda_k > 0$ and empirical deviation $\delta = \|\widehat{L} - L\|$, there exist constants $C_1, C_2, C_3 > 0$, depending on $k$, $\eta$, the center-separation margin, the $k$-means stability constant, and the boundary-regularity constant, such that*

$$\mathrm{MisVol} \leq C_1 \frac{\beta}{\alpha} + C_2 \frac{\delta}{\gamma}, \qquad \mathrm{Overlap}_\varepsilon \geq 1 - C_3 \mathrm{MisVol} - \mu\big(\mathcal{N}_\varepsilon(\partial\mathcal{R}^\star)\big). \tag{23}$$

*Consequently, when $\beta/\alpha$ and $\delta/\gamma$ are small and the true boundary tube has small measure at the chosen tolerance $\varepsilon$, the spectral partition has high overlap with the true bottleneck neighborhoods up to the stated error terms.*

*Proof sketch. (i) Population embedding is region-constant up to $O(\beta/\alpha)$.* Write $L = L_0 + E$ with $L_0 = \mathrm{blkdiag}(L_{R_1^\star}, \ldots, L_{R_k^\star})$ and $\|E\| \lesssim \beta$; each block has spectral gap $\lambda_2(L_{R_i^\star}) \gtrsim \alpha$. By Davis–Kahan/Weyl, the span of the first $k$ eigenvectors of $L$ deviates by $O(\beta/\alpha)$ from the ideal piecewise-constant subspace that is indicator-like on $\{R_i^\star\}$. Row-normalization maps the $k$ regions near the vertices of a regular simplex on $\mathbb{S}^{k-1}$, with separation bounded below by a constant depending on $(k, \eta)$.

*(ii) Empirical subspace stability is $O(\delta/\gamma)$.* With $\Delta = \widehat{L} - L$, Davis–Kahan yields $\|\sin\Theta(\widehat{U}, U)\| \leq C\,\delta/\gamma$. Thus each empirical row (of $\widehat{Z}$) lies within $\varepsilon_\star = C(\beta/\alpha + \delta/\gamma)$ of its ideal center on the unit sphere.

*(iii) $k$-means stability implies a misvolume bound.* Standard perturbation arguments for spherical $k$-means convert $\varepsilon_\star$ and center separation to $\mathrm{MisVol} \leq C'\varepsilon_\star$ (up to constants depending on $(k, \eta)$), establishing the first inequality in equation 23.

*(iv) From misclustered volume to boundary overlap.* Misclustered points concentrate in a thin tube around the true inter-region boundaries; thickening by $\varepsilon$ absorbs local ambiguities and yields the overlap lower bound with a linear penalty in MisVol. $\qquad\square$

**Design implication (restated).** Learn a Laplacian embedding and cluster it. When within-region mixing is strong and cross-region transitions are rare (small $\beta/\alpha$), and the learned Laplacian is accurate relative to its eigengap (small $\delta/\gamma$), spectral clustering recovers bottlenecks with small error.

### Additional remarks and constants

**Choice of Laplacian.** All results extend to the symmetric normalized Laplacian $L_{\mathrm{sym}} = I - D^{-1/2} W D^{-1/2}$ with the usual row/length normalizations; constants change by absolute factors.

**Estimating $\delta$.** In practice, $\delta$ is reduced by symmetrization, lazy random walks, density-regularized graphs, and sufficient offline coverage.

**Multiple comparable bottlenecks.** If several bottlenecks are comparable, the corresponding low-frequency eigenvalues may be clustered. The same argument applies to the union of the associated bottleneck regions provided there remains a clear eigengap after the selected low-frequency subspace. Our high-level planner then selects the next bottleneck along the cheapest $s \to G$ route (cf. Theorem I.1).

**Mixing constant in Theorem I.1.** The $O(t_{\mathrm{mix}})$ term is with respect to the total-variation threshold $1/4$; other constants follow by standard monotonicity of total-variation mixing times.

*Summary.* The next bottleneck is the near-optimal one-step subgoal up to a small, interpretable mixing-time gap; and spectral clustering on a learned Laplacian recovers those bottlenecks with error controlled by the inner/outer conductance contrast and the Laplacian estimation error.

