# OpenReview forum: "Bottleneck-Guided Spectral Subgoals For Offline Goal-Conditioned RL"
_ICML.cc/2026/Conference — ICML 2026 regular_

### Official Review · Reviewer_Gmny · 2026-03-02

**Soundness:** 3
**Presentation:** 3
**Significance:** 2
**Originality:** 3
**Overall Recommendation:** 3
**Confidence:** 4

**Summary:**

This paper reframes offline goal-conditioned reinforcement learning (OGCRL) as a routing problem across metastable regions connected by a small number of hard-to-cross bottlenecks. The authors propose the principle that the near-optimal subgoal in long-horizon tasks is the next bottleneck. To operationalize this idea, they learn a Laplacian representation from offline data and apply spectral clustering to identify bottlenecks, extracting keypoints (KPs) at boundary crossings. A directed KP reachability graph is then constructed, and high-level planning is performed via lightweight BFS over this graph. A pluggable low-level controller, implemented as either a diffusion-based planner or a lightweight MLP, drives the agent into each KP’s acceptance region. The paper provides theoretical results supporting bottleneck-guided subgoal optimality and spectral recovery of bottlenecks. Experiments on D4RL and OGBench demonstrate good performance across diverse navigation and manipulation tasks.

**Compliance With Llm Reviewing Policy:**

Affirmed.

**Final Justification:**

Thank you to the authors for the rebuttal and for the effort in addressing the comments. Nevertheless, I still find the definition of IΔ somewhat heuristic. Also, the relatively weak results in visual environments and the evaluation setup, which compares against HIQL rather than more recent graph-based methods, lead me to maintain my current score.

**Key Questions For Authors:**

## 1. Definition of Bottlenecks in Manipulation

* While bottlenecks are topologically clear in navigation tasks, it is less obvious whether manipulation environments (e.g., FrankaKitchen) exhibit the same metastable structure assumed in the theory.

* How are bottlenecks defined in manipulation tasks?

* Which state variables determine region boundaries?

* What empirical evidence supports that the metastable assumptions (fast intra-region mixing, low inter-region conductance) hold in these environments?


## 2. Stability and Sensitivity of KP Slice (IΔ) Selection

* The definition of KP as partial state coordinates (IΔ, vΔ) appears to be a central design choice.

* How sensitive is the selection of IΔ to dataset noise, representation learning errors, or clustering variations?

* Has the method been evaluated under different slice selection strategies?

* Is there any stability or ablation analysis for this component?


## 3. Extension to Visual-based Environments

* The experiments are conducted in state-based (numeric) environments.

* How can bottleneck discovery and KP definition be generalized to visual-only environments?

* Is there any experimental or architectural evidence that spectral clustering in learned representation space can reliably recover bottlenecks?


## 4. Comparison with Recent Graph-Based HRL (e.g., GAS, ICML 2025)

* Recent graph-based hierarchical RL methods such as GAS (ICML 2025) pretrains temporal-distance representations and select subgoals aligned with optimal movement dynamics.

* In datasets with short trajectories (e.g., antmaze-giant-stitch) and noisy actions & random exploration (e.g., antmaze-large-explore), the reported performance appears lower than GAS.


## 5. Sensitivity to Dataset Quality and Distribution

* The theoretical guarantees rely on metastable assumptions, which may depend on dataset coverage and transition statistics.

* How does the method behave when the dataset violates the metastable structure (e.g., poor coverage, high noise, uneven transition frequencies)?

* Is the approach robust to low-quality or biased offline data?

* Understanding the failure modes under different dataset regimes would clarify the practical reliability of the method.


## 6. KP in Partial Coordinates vs. Latent Subgoal Representations

* The KP graph transfer (G1) and controller transfer (G2) results are interesting. However, these gains may partly stem from defining KP in explicit state coordinates (e.g., global x–y positions).

* In numeric environments, using explicit global coordinates as subgoals may inherently simplify planning compared to latent-state subgoals used in methods such as HIQL or GAS.

* Have the authors evaluated a variant where KP is defined purely in latent representation space instead of partial state coordinates?

**Limitations:**

While the paper briefly mentions certain assumptions and scope considerations, the discussion of limitations appears somewhat limited and could be strengthened.

In particular, the following aspects would benefit from clearer acknowledgment and analysis:

## 1. Dependence on the Metastable Assumption

* The theoretical guarantees rely heavily on the metastable assumption (fast intra-region mixing and low inter-region conductance). The paper does not sufficiently discuss potential failure modes when these conditions do not hold. A clearer characterization of environments where the assumptions may break down would improve the rigor and transparency of the work.

## 2. Limited Scope of Applicability

* All experiments are conducted in state-based environments. The paper does not clearly discuss the feasibility or limitations of extending the approach to visual or high-dimensional representation-based settings. Clarifying this would help delineate the practical scope of the method.

## 3. Sensitivity to Dataset Quality

* Since the method relies on recovering bottlenecks from offline data, it may be sensitive to dataset coverage, noise, or biased transition distributions. The paper does not provide an analysis of robustness under low-quality or uneven datasets, which could affect the reliability of bottleneck discovery.

## 4. Effect of KP Definition (Coordinate vs. Latent Representation)

* The KP definition leverages explicit state coordinates (e.g., x–y positions in navigation). It would be informative to analyze performance when KP is defined purely in latent representation space rather than privileged coordinate subsets. This would clarify whether the gains arise from the bottleneck structure itself or from the use of explicit coordinate information.

## 5. Comparison with Recent Graph-Based Baselines

* A direct comparison with recent graph-based hierarchical RL methods (e.g., GAS) would strengthen the positioning of the work. Including either empirical comparisons or a more detailed conceptual differentiation would better justify the claimed novelty and practical advantage.

**Strengths And Weaknesses:**

## 1. Soundness

(1) Strengths

* The paper clearly justifies the use of bottlenecks as one-step subgoals through a hitting-time–based theoretical analysis.

* It provides a principled explanation of why Laplacian spectral clustering can recover bottlenecks under metastable assumptions.

* The empirical evaluation spans diverse navigation and manipulation benchmarks (D4RL, OGBench), and the experimental setup appears reasonably thorough, with consistent performance improvements reported across tasks.

(2) Weaknesses

* The theoretical analysis relies heavily on the metastable assumption (fast intra-region mixing and low conductance across boundaries), yet there is limited empirical evidence demonstrating how well this assumption holds in the tested environments.

* It is unclear whether bottlenecks in manipulation tasks satisfy the same structural assumptions as navigation tasks; the theoretical alignment between the metastable analysis and manipulation domains is not sufficiently discussed.

* The definition of KP slices (i.e., the selection of IΔ) appears heuristic in practice, and the paper does not provide a detailed stability or sensitivity analysis of this component.

## 2. Presentation

(1) Strengths

* The paper is consistently framed around the central idea that “OGCRL can be viewed as routing across bottlenecks,” which provides a coherent and easy-to-follow narrative.

* Visualizations (e.g., Fig. 3) effectively illustrate how spectral clustering identifies bottlenecks and how KPs align with low-conductance regions.

(2) Weaknesses

* The Laplacian theory section, while concise, abstracts away several implementation details, making it difficult for readers to fully understand the connection between spectral conditions and practical deployment.

* The automatic selection of KP slices (IΔ, vΔ) is not explained in sufficient detail, raising potential concerns about reproducibility.

* The discussion of experimental results is largely centered on navigation tasks, and the distinction between navigation and manipulation domains is not analyzed in depth, leaving the scope of applicability somewhat ambiguous.

## 3. Significance

(1) Strengths

* The paper introduces the perspective of bottleneck-guided subgoal selection for long-horizon sparse-reward problems, which is conceptually meaningful.

* By contrasting time-based hierarchical RL with topology-based decomposition, the work suggests a potentially useful direction for long-horizon planning research.

(2) Weaknesses

* The applicability of the method appears primarily tailored to navigation-style environments with clear topological structure, which may limit its broader impact across general RL settings.

* While several baselines have demonstrated results in visual environments, this paper evaluates only in state-based (numeric) environments, leaving its generalization to visual settings unverified.

* Although improvements are observed, it remains unclear whether the gains represent a paradigm-shifting advance. In particular, compared to recent related work (e.g., GAS, ICML 2025), the reported performance appears comparatively lower.

## 4. Originality

(1) Strengths

* Interpreting bottlenecks as hitting-time–optimal subgoals provides a novel theoretical perspective that distinguishes this work from prior hierarchical RL approaches.

(2) Weaknesses

* Core components such as Laplacian representation learning and spectral clustering have been extensively studied in prior work. As a result, the methodological contribution may be viewed as an incremental integration rather than a fundamentally new algorithmic framework.

---

> ### Author Rebuttal · Authors · 2026-03-31
>
> ## 1.Stability of $I_\Delta$ & bottlenecks in manipulation tasks
> $I_\Delta$ is not hand-crafted.
> (1) For each state dimension, we measure its temporal smoothness on the offline dataset (magnitude and continuity of $\Delta s_t$ along trajectories).
> (2) Dimensions with low variation and strong continuity are kept, while high-frequency noisy dimensions are discarded.
> (3) This procedure is fully automatic, making KP discovery reproducible and robust.
>
> Under this definition, the metastable assumption **matches manipulation tasks even better than navigation**. In Kitchen, the automatically selected $I_\Delta$ naturally retains low-frequency, task-relevant object-state dimensions, such as the on/off state of the light. **Changes in these dimensions mark the completion of subtasks and shifts in the agent’s behavior mode**. These dimensions are typically piecewise stable along trajectories and change only at a small number of critical transitions. As a result, BASS can identify bottlenecks clearly, such as moving the kettle or turning on the light.
>
> The results on Kitchen in **Table 1** also support this point:
>
> |Env|HIQL|Ours|
> |-|-:|-:|
> |kitchen-partial|65.0|83.3±4.9|
> |kitchen-mixed|67.7|86.0±2.8|
>
> ## 2.Can the method generalize to visual environments, and can bottlenecks still work in learned representation space?
>
> We evaluate on the visual-antmaze environments in **Table 2**. The results outperform baselines such as HIQL:
>
> |Env|Ours|HIQL|
> |-|-:|-:|
> |visual-antmaze-large-navigate|65.3±3.1|53±9|
> |visual-antmaze-large-stitch|39.3±3.1|28±2|
>
> ## 3.How does the method compare to recent graph-based OGCRL methods such as GAS?
>
> Although GAS is also graph-based, its objective is fundamentally different from ours. GAS builds a dense graph with thousands of ordinary waypoints, mainly for fine-grained planning, whereas BASS discovers a sparse graph of bottlenecks (about 10–50 keypoints), each corresponding to an interpretable decision point where behavior modes switch. Thus, GAS focuses on micro-level planning, while BASS provides macro-level guidance.
>
> Empirically, on shared benchmarks BASS achieves competitive performance with GAS in manipulation tasks such as kitchen-partial, and on navigation tasks our scores are close to the reported GAS results when its graph is constructed without aggressive data filtering.
>
> The remaining gap mainly comes from two design choices:
>
> (i) GAS uses a very dense graph and shortest-path planning, which substantially lowers the difficulty of navigation tasks; and (ii) GAS strongly optimizes the low-level layer via reward shaping and aggressive data filtering (retaining only 2–8% of high-quality transitions), while BASS intentionally uses simple plug-and-play low-level controllers without extra data curation or intrinsic rewards.This is because the core contribution of BASS is to **isolate the value of bottleneck-based subgoal selection, rather than to engineer the strongest possible low-level controller**.
>
> We therefore view BASS and GAS as complementary: GAS supports fine-grained waypoint planning, while BASS identifies the macro bottlenecks where agents must switch modes. A natural future direction is to combine both—using BASS bottlenecks as mandatory macro waypoints and GAS to refine micro-level paths within regions.
>
> ## 4.How robust is the method to low-quality offline datasets?
> We have already tested BASS on two representative types of low-quality offline data in **Table 2**). The first is antmaze-large-explore-v0, generated by a low-quality rule-based policy, which evaluates robustness to noisy data. The second is the family of stitch datasets (PointMaze / AntMaze / HumanoidMaze), which consist of short trajectory fragments and test the agent’s ability to perform long-horizon stitching.
> |Env|HIQL|Ours|
> |-|-:|-:|
> |pointmaze-large-stitch|13±6|99.3±1.2|
> |pointmaze-giant-stitch|0±0|85.3±3.1|
> |pointmaze-teleport-stitch|34±4|42.0±14.0|
> |antmaze-large-stitch|67±5|81.0±7.0|
> |antmaze-giant-stitch|2±2|71.3±7.0|
> |antmaze-large-explore|4±5|72.7±1.2|
> |humanoidmaze-large-stitch|28±3|45.3±3.1|
> |humanoidmaze-giant-stitch|3±2|55.3±3.1|
>
> ## 5.To what extent do the gains come from bottleneck-based subgoal discovery itself, rather than from using explicit state coordinates instead of latent subgoal representations?
> (1) The low-level controller is not conditioned only on explicit $(x,y)$-style coordinates; it always receives the full underlying state (or a latent embedding in visual settings). Explicit low-frequency coordinates are used only at the high level for Laplacian construction and reusable KP semantics.
>
> (2) The same advantage appears in visual-AntMaze, where BASS still significantly outperforms HIQL.
>
> (3) Under a shared Laplacian representation, the ALLO-centroid ablation shows that replacing bottleneck keypoints with cluster-centroid subgoals causes a large drop **(81.0 → 40.7 on antmaze-large-stitch)**. This indicates that the main gain comes from boundary-aligned bottleneck subgoals.

---

> > ### Author Rebuttal · Reviewer_Gmny · 2026-04-02
> >
> > Thank you to the authors for the rebuttal and for the effort in addressing the comments. Nevertheless, I still find the definition of IΔ somewhat heuristic. Also, the relatively weak results in visual environments and the evaluation setup, which compares against HIQL rather than more recent graph-based methods, lead me to maintain my current score.

---

### Official Review · Reviewer_CNMs · 2026-03-05

**Soundness:** 3
**Presentation:** 2
**Significance:** 3
**Originality:** 3
**Overall Recommendation:** 4
**Confidence:** 4

**Summary:**

This paper proposes a method to address the planning problem in long-horizon reinforcement learning (RL) tasks. By identifying Keypoints (KPs) from offline datasets and leveraging them for hierarchical planning, the method effectively solves the core challenge of long-horizon goal-conditioned offline RL (OGCRL). Empirically, the proposed approach outperforms existing methods across multiple benchmarks. However, all experiments are conducted in simple numerical environments, and several critical issues in the experiments lack clear explanations.

**Compliance With Llm Reviewing Policy:**

Affirmed.

**Final Justification:**

During the review of this paper, I focused on the generalization of the proposed method, specifically how KPs can be transferred from one task to another. I believe the authors have provided convincing explanations for this issue and resolved my confusion.

The paper presents thorough and clear theoretical proofs to support the proposed method. The approach is distinctive and addresses the bottleneck problem in offline goal-conditioned RL from a novel perspective. The proposed method is promising for applications in popular research fields such as robotics and carries strong academic value.

**Key Questions For Authors:**

No key questions.

**Limitations:**

Yes

**Strengths And Weaknesses:**

Strength:
1.The proposed method for identifying KPs from offline data is innovative. For scenarios where only offline data is available and scene information is hard to obtain, this approach can effectively extract KPs and apply them to path planning, filling the gap of topology-aware subgoal selection in OGCRL.
2.The theoretical analysis and explanations are comprehensive.
3.The experimental results strongly support the proposed ideas and methods.

Weakness:
1.The paper’s focus and innovation alignment need clarification. The core innovation lies in KP identification for high-level planning, which is decoupled from the RL algorithm itself. In the low-level controller section of the Method chapter, the paper mentions that Decision Diffuser or lightweight MLP can be used—actually, other non-RL methods could also be applied here. The authors should re-clarify the problem they aim to solve and their core innovations to better align with the actual work.
2.Lack of comparative experiments with alternative dimensionality reduction or clustering methods. The paper’s key innovation relies on Laplacian spectral clustering for bottleneck discovery. However, it does not compare with other common dimensionality reduction (e.g., PCA) or clustering methods to verify whether Laplacian spectral clustering is uniquely effective. Supplementary experiments with these alternatives are necessary to highlight the superiority of the proposed clustering strategy and strengthen the justification for the core innovation.
3.The generalization experiments are counterintuitive and insufficiently explained. According to the paper’s description, KPs trained from offline data are directly transferred to other environments with similar topology for planning. Intuitively, this is problematic—just as path points from Map A cannot be directly applied to Map B, even minor differences between environments may render direct KP transfer infeasible.

---

> ### Author Rebuttal · Authors · 2026-03-30
>
> We thank the reviewer for the constructive questions. We would like to first address W2.
>
> ## W2. Why is Laplacian spectral clustering necessary?
>
> In fact, **“Laplacian-based representation” and “spectral clustering partition” are not two separable modules**. By definition, **spectral clustering is clustering based on Laplacian representations**, rather than a particular clustering algorithm. In implementation, following the standard spectral clustering framework recommended by von Luxburg(2007) and Ng, Jordan, and Weiss(2002), we apply $k$-means to the row vectors of the first $d$ eigenvectors. Table 6 shows that performance is not sensitive to the choice of $k$:
>
> |Env|32|36|40|44|48|52|56|
> |-|-:|-:|-:|-:|-:|-:|-:|
> |antmaze-giant-stitch|13±2|11±3|53±3|60±9|68±4|63±2|67±5|
> |pointmaze-giant-stitch|85±1|87±3|81±1|85±5|89±2|83±6|89±6|
>
> |Env|10|15|20|24|32|36|40|44|48|
> |-|-:|-:|-:|-:|-:|-:|-:|-:|-:|
> |antmaze-large-play|32±4|15±7|96±2|95±3|95±2|92±2|91±3|94±3|90±3|
> |pointmaze-large-stitch|98±2|100±0|100±0|99±1|100±0|100±0|100±0|100±0|100±0|
>
> These results show a broad plateau of K values where performance remains stable and often matches or even exceeds the main-table results.
> Thus, what matters is not $k$-means per se, but the Laplacian representation: **unlike standard geometric clustering, it organizes states by connectivity and reachability, which is exactly what allows bottlenecks to emerge**.
>
> As stated in Appendix G, the representation dimension is only 20, so no extra dimensionality reduction is used. On antmaze-large-stitch, we further tested alternative clustering and dimensionality-reduction methods. The results show that distance-based clustering methods such as HDBSCAN performs slightly worse than K-Means (the standard choice in the spectral clustering), while BASS remains robust with PCA.
>
> |Method|Score|
> |-|-:|
> |HIQL|67 ± 5|
> |BASS (K-Means)|81.0 ± 7.0|
> |BASS (K-Means + PCA-5)| 80.7 ± 3.1 |
> |BASS (K-Means + PCA-10)| 82.0 ± 2.0 |
> |BASS (K-Means + PCA-15)| 80.7 ± 5.0 |
> |BASS (HDBSCAN)| 74.7 ± 3.1 |
>
> ## W1. What is the paper’s core innovation, and how is it aligned with RL?
>
> Our method is not merely a decoupled planning module. Both hierarchy levels are closely tied to RL. At the high level, keypoint discovery relies on Laplacian representation, which was **originally developed for RL to capture reward-free reachability structure**; spectral clustering is tightly coupled with this representation. At the low level, our pluggable controllers, Decision Diffuser and the MLP trained with a HIQL-style objective, are both **standard offline RL components**. In contrast, non-RL alternatives such as imitation learning are generally much less robust on low-quality offline data.
>
> More importantly, as an offline goal-conditioned RL (OGCRL) paper, our core contribution is **rethinking subgoal selection in the offline setting**. Subgoal selection is one of the most central and widely studied problems in OGCRL. Prior OGCRL methods typically use heuristic subgoals, such as midway states(Imagined Subgoals, ICML’21), fixed/skip-step future states(HIQL, NeurIPS’23; Simple Hierarchical Planning with Diffusion, ICLR’24; DiffuserLite, NeurIPS’24), fixed-horizon skills(OPAL, ICLR’21; SSD, AAAI’24), or highest-value states in short windows(ESD, AAMAS’25).
>
> These choices often lack semantic grounding: they do not explicitly identify hard-to-pass bottlenecks or the states. As a result, failures often occur exactly there, e.g., AntMaze turns or Kitchen grasp/transport transitions.
>
> Our key point is that **in OGCRL, subgoals should not only guide short-term motion, but also anticipate how to cross the next bottleneck and when to switch behavior modes**. To the best of our knowledge, **this is the first OGCRL work that explicitly formulates hard-to-cross bottlenecks in a metastable state space as subgoals**. We operationalize this by using Laplacian representations as a tool: we apply spectral clustering to offline data, expose metastable regions and boundaries, treat boundary crossings as keypoints, and build a directed keypoint graph for planning.
>
> ## W3. Why should the transferred keypoint graph generalize across environments?
>
> In the transfer experiments, the goal is diagnostic: to test whether the graph captures reusable structure rather than to claim universal transfer. If the graph merely memorized dataset-specific behaviors, performance should drop sharply after transfer. Instead, much of the performance is preserved, suggesting that it mainly captures topological bottleneck structure.
>
> The Point-to-Ant results further suggest that such structure may be easier to recover under simpler dynamics, which points to a promising future direction, this also points to a promising future direction: learning bottleneck structure in simpler systems and transferring it to more complex ones with the same topology, such as learning a keypoint graph with a two-finger gripper and reusing it as a topological prior for a five-finger hand.

---

> > ### Author Rebuttal · Reviewer_CNMs · 2026-04-04
> >
> > The authors have provided a detailed and thorough response to the questions I raised.
> > Regarding the transfer experiments, I recommend that the authors explicitly elaborate on the conditions for valid transfer in the revised manuscript—specifically, clarifying what environments that enable effective KP graph transfer. Such clarification will provide clear practical guidance for researchers intending to apply the proposed method to transfer scenarios.

---

> > > ### Author Response · Authors · 2026-04-07
> > >
> > > We thank the reviewer for this helpful follow-up. To make the transfer setting more precise, it is useful to distinguish between **whether KP-graph transfer is well-defined** and **whether it is likely to be effective in practice**. For clarity, we refer to the state dimensions in common offline GCRL tasks as consisting of **low-frequency dimensions** and **high-frequency dimensions**.
> > >
> > > **Low-frequency dimensions** characterize the global topological structure of the task and typically change slowly along trajectories, whereas **high-frequency dimensions** capture fast local control dynamics and usually vary much more rapidly. This distinction is quite general. For example, in navigation tasks, the agent’s position coordinates are low-frequency dimensions, while its velocity, joint states, and related motion variables are high-frequency dimensions. In manipulation tasks, the states of the target objects can be treated as low-frequency dimensions, whereas the robot arm’s pose and velocity belong to high-frequency dimensions.
> > >
> > > Under this distinction, KP-graph transfer first requires the following **transferability condition**:
> > >
> > > **(1) The source and target environments should share the same low-frequency topology within the same domain.**
> > >
> > > In other words, the region connectivity and bottleneck structure induced by the low-frequency state space should remain the same across the two environments. Under this condition, bottlenecks identified from the low-frequency dimensions retain the same semantic meaning and can therefore be transferred directly.
> > > For example, in our G1 experiment, PointMaze-Large can transfer to AntMaze-Large because the two environments share the same low-frequency state space: the maze layout is identical, and the xy coordinates have the same semantics. As a result, bottlenecks defined by xy positions can be directly transferred from PointMaze to AntMaze.
> > >
> > > Once this transferability condition is satisfied, achieving not only feasible but also **effective** transfer typically requires an additional condition:
> > >
> > > **(2) The source environment usually has lower high-frequency dynamical complexity than the target environment.**
> > >
> > > A possible explanation is that an environment with simpler high-frequency dynamics can explore the same low-frequency topological space more thoroughly, resulting in more uniform data coverage. In such cases, the reachability structure learned by the Laplacian representation is typically smoother, less noisy, and more accurate, which in turn leads to more reliable bottleneck identification. Consequently, a KP graph learned in a simpler environment can serve as a stronger topological prior when transferred to a target environment that shares the same low-frequency topology but has more complex high-frequency dynamics.
> > > This may help explain why PointMaze-Large → AntMaze-Large works in our experiments.
> > >
> > > In summary, **low-frequency topological consistency determines whether KP-graph transfer is semantically well-defined, while the relative complexity of the high-frequency dynamics affects whether such transfer is likely to be effective in practice.**
> > >
> > > We will incorporate this constructive distinction, together with the two conditions above, explicitly into the revised manuscript, both in the transfer experiment section and in the limitations discussion.
> > >
> > > We thank the reviewer again for the time and effort devoted to reviewing our paper. If the reviewer finds that our clarification has adequately addressed the remaining concern, we would sincerely appreciate reconsideration of the score.

---

### Official Review · Reviewer_QQqx · 2026-03-11

**Soundness:** 2
**Presentation:** 2
**Significance:** 2
**Originality:** 2
**Overall Recommendation:** 4
**Confidence:** 4

**Summary:**

This paper proposes BASS, a bottleneck-finding algorithm that extracts keypoints from offline data and organizes them into a planning graph to enable hierarchical planning. Theoretical analysis shows that routing through bottlenecks yields an approximately optimal one-step subgoal and that the Laplacian spectrum can effectively capture these bottlenecks. Empirical results demonstrate the effectiveness of the proposed method in environments such as AntMaze and Kitchen.

**Compliance With Llm Reviewing Policy:**

Affirmed.

**Final Justification:**

My questions were fully addressed

**Key Questions For Authors:**

Weaknesses are also my questions.

**Limitations:**

No, the authors didn't adequately discuss the limitations and potential negative societal impact of their work. A brief analysis of the limitations would be appreciated.

**Strengths And Weaknesses:**

Strengths:

1. The framework is clearly described, making it easy to understand both its overall design and the role of each component. The key methods are thoroughly explained.

2. The theoretical discussion of policy optimality and the coverage of spectral clustering is solid.

Weaknesses:

1. BASS comprehensively suffers from common problems in offline learning. Some key questions, such as how to handle noisy offline data and how to bridge the gap between offline data and online deployment, require further discussion.

2. BASS introduces a keypoint finding and planning process, which could significantly increase the computational load. Therefore, I suggest adding a time analysis to justify the choice of BASS over other baselines in terms of performance-time consumption balance.

---

> ### Author Rebuttal · Authors · 2026-03-31
>
> ## W1. BASS comprehensively suffers from common problems in offline learning. Some key questions, such as how to handle noisy offline data and how to bridge the gap between offline data and online deployment, require further discussion.
>
> Learning from low-quality offline data is indeed a central challenge in offline RL. To address this issue, we have already considered two types of low-quality data in our experiments.
>
> The first is **antmaze-large-explore-v0**, a dataset generated by a low-quality rule-based policy, which evaluates the ability to learn from noisy or suboptimal data.
>
> The second consists of the **pointmaze / antmaze / humanoid-stitch** datasets, which are composed of short trajectory segments and therefore test the agent’s ability to perform effective trajectory stitching. Empirically, our method significantly outperforms mainstream offline GCRL baselines such as HIQL on these settings, as shown in **Table 2** of our paper.
> | Dataset Group | Environment | GCBC | GCIVL | GCIQL | QRL | CRL | HIQL | BASS (Ours) |
> |---|---|---:|---:|---:|---:|---:|---:|---:|
> | PointMaze | pointmaze-large-stitch-v0 | 7 ± 5 | 12 ± 6 | 31 ± 2 | 84 ± 15 | 0 ± 0 | 13 ± 6 | **99.3 ± 1.2** |
> | PointMaze | pointmaze-giant-stitch-v0 | 0 ± 0 | 0 ± 0 | 0 ± 0 | 50 ± 8 | 0 ± 0 | 0 ± 0 | **85.3 ± 3.1** |
> | PointMaze | pointmaze-teleport-stitch-v0 | 31 ± 9 | 44 ± 2 | 25 ± 3 | 9 ± 5 | 4 ± 3 | 34 ± 4 | **42.0 ± 14.0** |
> | AntMaze (OGBench variants) | antmaze-large-stitch-v0 | 3 ± 3 | 18 ± 2 | 7 ± 2 | 18 ± 2 | 11 ± 2 | 67 ± 5 | **81.0 ± 7.0** |
> | AntMaze (OGBench variants) | antmaze-giant-stitch-v0 | 0 ± 0 | 0 ± 0 | 0 ± 0 | 0 ± 0 | 0 ± 0 | 2 ± 2 | **71.3 ± 7.0** |
> | AntMaze (OGBench variants) | antmaze-large-explore-v0 | 0 ± 0 | 10 ± 3 | 0 ± 0 | 0 ± 0 | 0 ± 0 | 4 ± 5 | **72.7 ± 1.2** |
> | HumanoidMaze | humanoidmaze-large-stitch-v0 | 6 ± 3 | 1 ± 1 | 0 ± 0 | 3 ± 1 | 4 ± 1 | 28 ± 3 | **45.3 ± 3.1** |
> | HumanoidMaze | humanoidmaze-giant-stitch-v0 | 0 ± 0 | 0 ± 0 | 0 ± 0 | 0 ± 0 | 0 ± 0 | 3 ± 2 | **55.3 ± 3.1** |
>
> In addition, like mainstream offline RL methods, **all results in this paper are evaluated in an online environment**, i.e., the policy is learned offline and then tested online to assess its offline-training-to-online-deployment capability.
>
> ## W2. I suggest adding a time analysis to justify the choice of BASS over other baselines in terms of performance-time consumption balance.
> Taking the **antmaze-giant** environment as an example, we conducted a timing study and compared our method with the baseline. All timing experiments were run on the same server with a single NVIDIA RTX 3090 GPU.
>
> At inference time, our method requires **3.2s** to complete a full rollout for one instance. The average latency per step is about **0.00585s**, corresponding to **170.9 Hz**. This feedback speed is comparable to methods such as HIQL, while being significantly faster than diffusion-based planners/policies, whose inference times in this setting are **0.1s-0.5s**, respectively.
>
> Compared with HIQL, the additional inference-time overhead in our method comes from keypoint finding and planning. In Appendix E, we analyze the time complexity of this procedure in detail and show that it can be completed efficiently with a BFS algorithm. In practice, this step takes only **0.002s**, accounting for approximately **0.63%** of the total inference time.
>
> As for training time, the extra overhead relative to HIQL comes from the Laplacian module. In our experiments, this additional cost consists of **597.7s** for training the Laplacian representation, **4.04s** for inferring the Laplacian representations of the offline dataset, and **3.44s** for performing spectral clustering on these representations, for a total of **605.2s**. This is much smaller than the **1.5–3 hours** required to train HIQL. It is also shorter than the training time of diffusion-based planners/policies, which is about **12h–48h**. We will add a new runtime comparison table in the revised version.
>
>
> It is also worth noting that the Laplacian representation implementation we used is trained in PyTorch, whereas HIQL is trained in JAX; if the Laplacian representation training were reimplemented in JAX, the training speed could be further improved.

---

> > ### Author Rebuttal · Reviewer_QQqx · 2026-04-04
> >
> > Thank you for your response. I appreciate the clarifications provided, which have fully addressed my questions. I am satisfied with the authors' rebuttal and will raise my score.

---

### Official Review · Reviewer_8ZLN · 2026-03-13

**Soundness:** 3
**Presentation:** 3
**Significance:** 2
**Originality:** 2
**Overall Recommendation:** 4
**Confidence:** 4

**Summary:**

The paper focuses on the offline reinforcement learning setting where the eigenvectors of a Laplacian is used to obtain clusters (via spectral clustering) where the boundaries of clusters are used as sub-goals for planning. The connection between bottlenecks and the boundaries of the clusters is explored and theoretically expressed. The proposed method is compared against relevant comparative baselines showcasing empirical gains against the non-Laplacian bases baselines.

**Compliance With Llm Reviewing Policy:**

Affirmed.

**Final Justification:**

The authors have resolved my questions. I have increased the soundness score from fair to good in light of the author's rebuttal and have changed my overall score from weak reject to weak accept.

**Key Questions For Authors:**

- Have the authors considered the performance if instead of the boundaries of clusters, the centers, or even random states in clusters were used as sub goals? I suspect in heavily bottlenecked domains the bottleneck based sub-goal could provide gains against the centers of clusters, however, in the maze-like domains evaluated, it's not clear why centers or any relatively intermediately distanced sub-goals approach still using the Laplacian representations would not be just as good.

- Have the authors considered conceptually the usecase in the online settings? Do the authors consider the method extendable to the online setting as it is or would the problem of uniform sampling have to be addressed?

**Limitations:**

yes

**Strengths And Weaknesses:**

Strengths:

- The method proposed, BASS, shows clear performance gains against relevant comparative approaches.
- The paper provides a theoretical contribution to show the optimality of the bottleneck-guided sub-goals.
- Generally the paper is well-written and the approach was clearly understandable from the writing and visualization figures.

Weaknesses:

- Is it not clear how much of the performance improvement above other comparative approaches, if any, is due to the use of the bottlenecks between spectral clustering partitions and how much is due to the underlying Laplacian representation learning.
- Whilst the theoretical details are good to see, the general use of bottlenecks as sub-goals is not new, neither is the relation to Laplacian representations in it's entirety.
- The results focus on the offline setting where Laplacian representations are learnable without as much of a problem in terms of the uniformity of the samples obtained - that is, in the online setting, the inner product and smoothness terms of the ALLO approach can be heavily skewed by non-uniform visitation which in the context of this spectral cluster approach in the paper may lead to significant clustering problems (such as clustering diversely only on highly visited regions where orthogonality is more greatly expressed).

---

> ### Author Rebuttal · Authors · 2026-03-30
>
> ## W1. Disentangling the effects of Laplacian representation learning and spectral clustering
> In fact, “Laplacian-based representation” and “spectral clustering partition” **are not two separable modules**. By definition, **spectral clustering is clustering based on Laplacian representations**, rather than a particular clustering algorithm.
>
> In implementation, following the standard spectral clustering framework recommended by von Luxburg (2007) and Ng, Jordan, and Weiss (2002), we apply $k$-means to the row vectors of the first $d$ eigenvectors. Table 6 further shows that performance is not sensitive to the choice of $k$.
>
> ## W2. The novelty of this work
> We do not claim that bottleneck subgoals are new in the broader HRL literature, nor that Laplacian representations are new in RL. In fact, our core contribution is not a new Laplacian RL algorithm; rather, as an offline goal-conditioned RL paper, we **rethink subgoal selection in the offline setting**.
> To the best of our knowledge, **this is the first work in offline goal-conditioned RL (OGCRL) that explicitly formulates hard-to-cross bottlenecks in a metastable state space as subgoals**.
>
> Most prior OGCRL methods use heuristic subgoals, e.g., by fixed time windows or short-horizon reachability. Such subgoals often lack semantic grounding: they do not explicitly identify hard-to-pass bottlenecks where behavior modes should switch. As a result, failure often occurs exactly there, e.g., AntMaze agents colliding or falling at turns, and Kitchen agents misaligning during grasping and transport. Our key point is that in OGCRL, **subgoals should not only guide short-term motion, but also anticipate how to cross the next bottleneck and when to switch behavior modes**. We operationalize this using Laplacian representations as a tool: we apply spectral clustering to offline data, expose metastable regions and boundaries, treat boundary crossings as keypoints, and build a directed keypoint graph.
>
> ## W3&Q2. On extending the method to the online setting
> Prior work suggests that online co-training of Laplacian representations and RL policies is feasible. For example, *Deep Laplacian-based Options for Temporally-Extended Exploration* learns Laplacian representations from the replay buffer while training the policy. More recently, *Online Laplacian-Based Representation Learning in RL* studies this setting explicitly and provides theoretical support for simultaneous online updates of representation and policy. These results make an online extension conceptually promising. That said, additional techniques would likely be needed to mitigate non-uniform visitation and occupancy bias in the learned Laplacian and spectral partitions. Since this paper focuses on OGCRL, we leave such extensions to future work.
>
> ## Q1. On bottleneck locations in maze-like environments and using alternative subgoals
>
> In maze-like domains, bottlenecks typically arise at intersections and turning points, i.e., states that many trajectories must pass through but that are relatively hard to traverse. Their traversal difficulty also depends on the agent dynamics. For example, in AntMaze, the ant is more likely to fail at turns than the point mass in PointMaze. Thus, bottleneck-aware subgoals should matter more for complex dynamics.
>
> To test this, we used the centroid of each spectral cluster as the subgoal and compared it against BASS.
>
> | Environment               | Centroid baseline (mean ± std, %) | BASS (ours, mean ± std, %) |
> | ------------------------- | --------------------------------: | -------------------------: |
> | antmaze-large-stitch   |                        40.7 ± 6.1 |                 81.0 ± 7.0 |
> | pointmaze-large-stitch |                        97.3 ± 1.2 |                 99.3 ± 1.2 |
>
> On antmaze-large-stitch-v0, BASS nearly doubles the success rate over the centroid baseline. On the simpler pointmaze-large-stitch-v0, the gap is much smaller.
>
> More broadly, regarding your question about “relatively intermediately distanced” subgoals, we also compared against QPHIL, a OGCRL method that learns a VQ-VAE-based quantization of the state space and plans over the resulting discrete zone-like tokens. In this sense, QPHIL provides a useful comparison to geometric or region-level intermediate subgoal planning.
>
> | Environment               | QPHIL  | BASS (ours) |
> | ------------------------- | -----: | ----------: |
> | antmaze-medium-play    | 91 ± 2 | 98.0 ± 0.0 |
> | antmaze-medium-diverse | 92 ± 4 | 96.7 ± 0.9 |
> | antmaze-large-play     | 80 ± 3 | 96.0 ± 1.6 |
> | antmaze-large-diverse  | 82 ± 6 | 98.7 ± 1.9 |
> | antmaze-ultra-play     | 62 ± 4 | 97.3 ± 0.9 |
> | antmaze-ultra-diverse  | 62 ± 7 | 88.0 ± 1.6 |
>
> From the experimental results, BASS consistently achieves higher success rates. This suggests that geometry-based intermediate subgoals still tend to miss the truly critical transition states. By contrast, our bottleneck-aligned keypoints explicitly identify where the agent must pass next.

---

> > ### Author Rebuttal · Reviewer_8ZLN · 2026-04-04
> >
> > I thank the authors for their rebuttal. Whilst I still have concerns on the originality and significance of the approach, I will raise my score from 3 to 4 (weak accept) as the rebuttal has addressed my concern on the soundness of the evaluations.

---

> > > ### Author Response · Authors · 2026-04-07
> > >
> > > We sincerely thank the reviewer for the careful reading of our paper and rebuttal, and especially for reconsidering the score. We are very encouraged that our clarification has addressed your concern on the soundness of the evaluations.
> > >
> > > We also appreciate your remaining comments on originality and significance. In the revised manuscript, we will further sharpen the positioning of our contribution and make the scope of novelty more explicit, particularly that our novelty lies in explicitly formulating bottleneck-guided subgoal selection in offline goal-conditioned RL and operationalizing it through offline Laplacian spectral partitioning. Thank you again for your time and constructive feedback, and for your positive reconsideration of our paper.

---

### Decision · Program_Chairs · 2026-04-30

**Decision:**

Accept (regular)

**Comment:**

The reviewers found this paper to present a clear and promising approach to subgoal selection in offline goal-conditioned RL. In particular, they viewed the bottleneck-guided perspective as meaningful, and the empirical results across navigation and manipulation tasks were generally strong.

The rebuttal addressed several important concerns by clarifying runtime overhead, transfer conditions, robustness on challenging offline datasets, and the role of bottleneck-aligned keypoints relative to simpler alternatives. At the same time, some limitations remain, including the heuristic nature of certain design choices, the need for clearer positioning relative to recent graph-based approaches, and a more explicit discussion of the scope under which transfer is expected to work.

Overall, this is a technically solid paper with useful empirical evidence and a well-motivated central idea, though some aspects of the framing and evaluation could still be strengthened. I recommend weak acceptance at this stage.